# Compromised base excision repair pathway in *Mycobacterium tuberculosis* imparts superior adaptability in the host

Saba Naz[1,2], Shruti Dabral[3], Sathya Narayanan Nagarajan[1¤a], Divya Arora[1¤b], Lakshya Veer Singh[3], Pradeep Kumar[4¤c], Yogendra Singh[2], Dhiraj Kumar[3], Umesh Varshney[4]*, Vinay Kumar Nandicoori[1]*

**1** Signal Transduction Lab, National Institute of Immunology, Aruna Asaf Ali Marg, New Delhi, India, **2** Department of Zoology, University of Delhi, Delhi, India, **3** Cellular Immunology Group, International Center for Genetic Engineering and Biotechnology, Aruna Asaf Ali Marg, New Delhi, India, **4** Department of Microbiology & Cell Biology, Indian Institute of Sciences, Bangalore, India

¤a Current address: Molecular Microbiology and Structural Biochemistry, UMR5086, University Claude Bernard Lyon 1, Center National de la Recherche Scientifique, Lyon, France
¤b Current address: Molecular Immunity Unit, Department of Medicine, University of Cambridge, Francis Crick Avenue, Cambridge Biomedical Campus, Cambridge, United Kingdom
¤c Current address: Division of Infectious Disease, Department of Medicine, Rutgers University, New Jersey Medical School, Newark, New Jersey, United States of America
* varshney@iisc.ac.in (UV); vinaykn@nii.ac.in (VKN)

**Data Availability Statement:** Raw data of sequenced strains are deposited to the NCBI under Bioproject PRJNA634239. Metadata is available at:

## Abstract

Tuberculosis caused by *Mycobacterium tuberculosis* (*Mtb*) is a significant public health concern, exacerbated by the emergence of drug-resistant TB. To combat the host's dynamic environment, *Mtb* encodes multiple DNA repair enzymes that play a critical role in maintaining genomic integrity. *Mtb* possesses a GC-rich genome, rendering it highly susceptible to cytosine deaminations, resulting in the occurrence of uracils in the DNA. UDGs encoded by *ung* and *udgB* initiate the repair; hence we investigated the biological impact of deleting UDGs in the adaptation of pathogen. We generated gene replacement mutants of uracil DNA glycosylases, individually (*RvΔung*, *RvΔudgB*) or together (*RvΔdKO*). The double KO mutant, *RvΔdKO* exhibited remarkably higher spontaneous mutation rate, in the presence of antibiotics. Interestingly, *RvΔdKO* showed higher survival rates in guinea pigs and accumulated large number of SNPs as revealed by whole-genome sequence analysis. Competition assays revealed the superior fitness of *RvΔdKO* over *Rv*, both in *ex vivo* and *in vivo* conditions. We propose that compromised DNA repair results in the accumulation of mutations, and a subset of these drives adaptation in the host. Importantly, this property allowed us to utilize *RvΔdKO* for the facile identification of drug targets.

## Author summary

Mutation in the genome of bacteria contributes to the acquisition of drug resistance. Mutations in bacteria can arise due to exposures to antibiotics, oxidative, reductive, and many other stresses that bacteria encounter in the host. *Mtb* has multiple DNA repair

https://www.ncbi.nlm.nih.gov/bioproject/PRJNA634239.

**Funding:** This work was funded by the Department of Biotechnology, Government of India (BT/PR13522/COE/34/27/2015) to VKN & UV; J.C Bose fellowship (JCB/2019/000015) to VKN & (SR/S2/JCB-63/2007) to UV provided by Department of Science and Technology, Ministry of Science and Technology. National Institute of Immunology (NII) Core funding to VKN. DBT-IISc partnership program, and DST-FIST level II funding to UV. The Jamsetji Tata Trust funding to UV. CSIR-Senior Research Fellowship to SN. SN is an SRF in JCB/2019/000015 fellowship of VKN. The funders had no role in study design, data collection and analysis, decision to publish, or preparation of the manuscript.

**Competing interests:** The authors have declared that no competing interests exist.

mechanisms, including a base excision repair pathway to restore the damaged genome. Here we set out to determine the impact of deleting the Uracil DNA base excision pathway on pathogen adaptability to both antibiotic and host induced stresses. Combinatorial mutant of *Mtb* UDGs showed higher spontaneous rates of mutations when subjected to antibiotic stress and showed higher survival levels in the guinea pig model of infection. Whole-genome sequence analysis showed significant accumulation of SNPs, suggesting that mutations providing survival advantage may have been positively selected. We also showed that double mutant of *Mtb* UDGs would be an excellent means to identify antibiotic targets in the bacteria. Competition experiments wherein we pitted wild type and double mutant against each other demonstrated that double mutant has a decisive edge over the wild type. Together, data suggest that the absence of a base excision repair pathway leads to higher mutations and provides a survival advantage under stress. They could be an invaluable tool for identifying targets of new antibiotics.

## Introduction

The bacterium responsible for causing tuberculosis (TB) disease -*Mycobacterium tuberculosis* -is among the most notorious human pathogens prevalent across the world. Although multiple drugs are available to treat TB, the emergence of **m**ulti**d**rug-**r**esistant (MDR) and e**X**tensively **d**rug-**r**esistant (XDR) tuberculosis is a major cause of concern [1]. Almost half-a-million cases of MDR/Rifampicin resistant-TB were reported in 2019 [2]. While the success rate of treatment for drug-susceptible TB is ~85%, and only ~57% for MDR-TB cases. These numbers decline further for patients co-infected with HIV [2,3].

The acquisition of drug resistance in *Mtb* is not a simple mechanism. It is a conglomeration of genetic events that occur sequentially, described as a probable pre-resistance state that predisposes pathogen to eventual antibiotic resistance [4]. Drug resistance in *Mtb* can arise due to mutations in the direct targets of antibiotics or the drug-activating/modifying enzymes. Mutations in the regulatory regions of genes that cause overexpression of drug resistance-conferring genes, including but not restricted to efflux pumps, also help in decreasing the intracellular concentration of drugs. However, in 10–40% of clinical cases, drug resistance cannot be explained by mutations in the direct targets, suggesting hitherto unknown mechanisms employed by the bacilli [5]. Beijing family strains have a higher propensity to develop drug resistance [6]. It is speculated that the higher mutation rate of the Beijing strains could be due to polymorphisms in the genes involved in DNA repair, DNA replication, and recombination [7,8]. In fact, mutations in DNA repair/replication genes such as *dnaQ*, *alkA*, *nth* and *recF* have been identified in drug-resistant strains [9–11].

When *Mtb* enters the host cell, it encounters stress conditions in the form of reactive nitrogen and oxygen species, capable of damaging the nucleotides resulting in the spectrum of mutations in the genome and challenging the genomic integrity in bacteria [12–15]. *Mtb* genome is GC rich, which renders this pathogen more susceptible to cytosine deaminations resulting in C to U change. Inability or failure to correct such mutations prior to replication would result in the accumulation of CG to TA mutations in the genome [16]. Uracils may also arise in the genome because of their direct incorporation by DNA polymerases during replication [17]. Uracil DNA glycosylase (UDG), initiates the uracil excision repair pathway by catalyzing uracil removal from DNA. Analysis of the genome sequence of *Mtb* unveiled genes that encode proteins involved in uracil excision repair, namely Ung and UdgB, are reported to be non-essential for *in vitro* growth of *Mtb* [18,19] (https://mycobrowser.epfl.ch/). Ung belongs

to a family I of UDGs which act on the single and double-stranded DNA and are highly specific in recognizing and removing uracils whereas UdgB is a family V UDG that acts exclusively on double-stranded DNA. UdgB is a thermotolerant enzyme that excises uracil, hypoxanthine, and ethenocytosine from the double-stranded DNA. UdgB is an Fe-S cluster containing protein whose activity can be regulated by the availability of iron [20]. Previously we have shown that simultaneous deletion of *ung* and *udgB* in *Mycobacterium smegmatis* (*Msm*), showed a synergistic effect on the accumulation of mutations [21].

We hypothesized that deletion of DNA base excision repair genes may help bacteria in better adaptability under stress conditions. We adopted the route of generating gene replacement mutants for both the uracil DNA glycosylases (*ung* and *udgB*) individually (*RvΔung* and *RvΔudgB*) or together (*RvΔdKO*) to fast forward the *in vitro* and *in vivo* evolution of the pathogen. Based on the results presented here, we propose that deletion of DNA repair pathway genes translates into the higher accumulation of random mutations in the bacteria under stress. A selected subset of these mutations aids in the adaptation of the organism to varied selection pressure such as the host immune response.

## Methods

### Ethics statement

Animal experiments protocol was approved by the Animal Ethics Committee of the National Institute of Immunology, New Delhi, India. The approval (IAEC#409/16) is as per the guidelines issued by Committee for Control and Supervision of Experiments on Animals (CPCSEA), Government of India.

### Generation of gene replacement mutants and growth pattern analysis

Uracil DNA glycosylase gene replacement mutants were generated using a specialized transduction method. Briefly, upstream and downstream regions of *ung* were amplified using specific primers and the amplicons were cloned individually in pGEM-T vector. Upstream and downstream flanks were digested with appropriate restriction enzymes and subcloned in pYUB854 to generate pYUB854-*ung* [22]. pYUB854-*ung* was digested with PacI and ligated into phAE159 at the compatible PacI site to generate recombinant phAE159-*ung*. pYUB1471 was digested with PflMI to obtain 1.6 kb *oriE*+ λ *cos* sites DNA fragment. PCR amplicons were digested with appropriate restriction enzymes and ligated with *oriE*+λ *cos* and chloramphenicol resistance cassette to generate AES. AES was linearized and ligated with phAE159 in the compatible PacI site to generate recombinant phAE159-*udgB* [23]. Transduction was performed as described previously and colonies obtained after transduction were screened for recombination at the native locus by performing multiple PCR reactions. For *in vitro* growth measurements, *Rv*, *RvΔung*, *RvΔudgB*, and *RvΔdKO* were grown either in 7H9-ADC or Sauton's minimal medium upto $A_{600}$~0.6 and inoculated into a fresh medium at final of $A_{600}$~0.1 in technical quadruplet. CFUs were enumerated at day 0, 3, 6, and 9 on 7H11-OADC plates. Growth kinetics was performed using two independent biological experiments and each biological experiment was performed in quadruplet. Statistical analysis (one way ANOVA) was performed using n = 4 for each biological experiment. Graphpad Prism software was employed for performing statistical analysis.

### Preparation of whole-cell extract and western blotting

Cultures were inoculated in 7H9- ADC medium at $A_{600}$ ~0.1 and grown till $A_{600}$ ~0.8. Cells were centrifuged at 4000 rpm for 10 min at room temperature (RT) and resuspended in the

lysis buffer containing protease inhibitors. Cells were transferred into bead beating tubes containing zirconium beads and bead-beating was performed until cell lysis. Lysates were centrifuged twice at 13000 rpm for 45 min at 4˚C. The supernatant was transferred into a fresh Eppendorf tube and protein estimation was performed using Bradford assay. Cell lysates (10, 50, and 100 μg) of *Rv*, *RvΔung*, *RvΔudgB*, and *RvΔdKO* were loaded on 16% Tris-Tricine gel containing 6M urea and transferred to nitrocellulose membranes. Membranes were blocked using 5% BSA prepared in $1XPBST_{20}$ for 2 h, and incubated overnight at 4˚C with α-PknB (1:10000), α-Ung (1:2000), and α-UdgB (1:2000) antibodies, respectively. Membranes were washed thrice using $1XPBST_{20}$ and incubated with anti-rabbit secondary antibody, DARPO (1:10000) for 1h at RT, and washed thrice with $1XPBST_{20}$. Nitrocellulose membranes were incubated with chemiluminescence reagents and exposed to X-ray film and developed.

## UDG activity assay

DNA oligomers, SSU9 and GU9 (10 pmols) were 5' end-labeled using 10 μCi of γ-[$^{32}$P]ATP and T4 polynucleotide kinase. After labeling, oligomers were passed through Sephadex G50 mini-columns to remove free γ-[$^{32}$P]ATP, and the oligomers were eluted in 25 μl of 1X Ung assay buffer (20 mM Tris–HCl (pH 7.5), 100 mM NaCl, 10% glycerol and 2 mM β-mercaptoethanol). To determine Ung activity, 10 μg of *Rv*, *RvΔung*, *RvΔudgB*, and *RvΔdKO* cell lysates were incubated with radiolabeled SSU9 and GU9, in the presence of 1X Ung buffer and the reactions were incubated at 37˚C for 1 h. Reactions were stopped by adding 10 μl of 0.1 N NaOH, and the abasic sites were cleaved by heating at 90˚C for 30 min. Samples were vacuum dried and resuspended in 8 μl formamide dye. Samples were boiled for 10 min at 95˚C, centrifuged, and loaded on 15% polyacrylamide-8M urea gels. For assessing the activity of UdgB, 25 ng of Ugi was added to the lysates of *Rv*, *RvΔung*, *RvΔudgB*, and *RvΔdKO* and incubated for 30 min prior to performing the UDG assays. Gels were exposed overnight on X-ray film and an autoradiogram was developed after 24 h. Similar procedure was employed for performing Ung or UdgB activity assay using *Rv*, *RvΔung*, *RvΔung*::*ung*, *RvΔudgB*, *RvΔudgB*::*udgB*, *RvΔdKO* and *RvΔdKO*::*ung-udgB*.

## Generation of complementation constructs

*Ung* and *udgB* independently and together were cloned in pST-Ki vector [24]. *Ung* and *udgB* were PCR amplified using specific primers from the *Rv* genomic DNA. Amplicons were digested with NdeI and HindIII restriction enzymes and cloned into the corresponding sites in pST-Ki to generate pST-Ki-*ung* and pST-Ki-*udgB*. To generate a double complementation construct harboring both the *ung* and *udgB*, pST-Ki-*udgB* was digested with ScaI-HpaI, and pST-Ki-*ung* was digested with SnaBI and treated with Antarctic phosphatase. pSTKi-*ung* was ligated with *udgB* fragment to generate pST-Ki-*ung-udgB*.

## Analysis of mutation rates

Antibiotic sensitivity of *Rv*, *RvΔung*, *RvΔung*::*ung*, *RvΔudgB*, *RvΔudgB*::*udgB*, *RvΔdKO, and RvΔdKO*::*ung-udgB* was determined by spotting them on rifampicin, ciprofloxacin, isoniazid, and no antibiotic-containing plates. Colonies that were antibiotic sensitive were selected for spontaneous mutation rate analysis experiments. Three antibiotic-sensitive cultures of *Rv*, *RvΔung*, *RvΔudgB,RvΔdKO*, *RvΔung*::*ung*, *RvΔudgB*::*udgB* and *RvΔdKO*::*ung-udgB* were grown in 7H9-ADC medium to $A_{600}$ ~0.6 and cells were inoculated in 10 ml fresh medium (at 50,000 cells/ml)containing 15% sterile culture filtrate of H37Rv. Cultures were grown for 15 days by incubation at 37˚C at 200 rpm. On the 15$^{th}$ day, appropriate dilutions were plated on 7H11-OADC plates in the absence of antibiotic for enumerating the total number of bacteria,

while 1 ml was plated on either rifampicin-containing (2 μg/ml) or ciprofloxacin-containing (1.5 μg/ml) or isoniazid (10 μg/ml) 7H11-OADC for enumerating antibiotic-resistant colonies. Spontaneous mutation rates were calculated using David's fluctuation test [25,26]. For determination of mutation frequency under oxidative stress conditions, *Rv*, *RvΔung*, *RvΔung::ung*, *RvΔudgB*, *RvΔudgB::udgB*, *RvΔdKO*, and *RvΔdKO::ung-udgB* strains were inoculated at $A_{600}$ ~0.1 and grown to $A_{600}$ ~ 0.6 and 50 μM CHP added for 24 h. Subsequently, (24 h later) cells were collected by centrifugation and plated on no antibiotic or rifampicin-containing (10 μg/ml) medium for calculating the mutation frequency. For analysis under nitrosative stress, cells were incubated for 48 h in an acidic medium containing 6 mM sodium nitrite, at $A_{600}$ ~0.6. Mutation frequency was calculated by plating after 48 h post sodium nitrite addition on no-antibiotic or rifampicin-containing medium. Spontaneous mutation rate or mutation frequency was performed using two independent biological experiments and each biological experiment was performed in triplicates. Statistical analysis (one-way ANOVA) was performed using n = 3 or n = 6. Graphpad Prism software was employed for performing statistical analysis. Data represents one of the two biological experiments.

## Guinea pig infection

Cultures of *Rv*, *RvΔung*, *RvΔudgB*, *RvΔdKO*, and *RvΔdKO::ung-udgB* were grown to $A_{600}$ ~0.8. Cells were centrifuged at 4000 rpm for 10 min at room temperature, resuspended in saline, and passed through a 26½ gauge needle to ensure single-cell suspension. Cells ($1 \times 10^8$) were suspended in 15 ml saline for infection. Female outbred Hartley guinea pigs were challenged through an aerosol route using a Madison chamber calibrated to deliver ~100 bacilli/lung. No antibiotic treatment was given to guinea pigs during or after infection. CFUs were enumerated at day 1 post infection (p.i) for assessing the implantation of bacilli and 56-days p.i for assessing the survival in the lungs and spleen. Statistical analysis was performed at day 1 using n = 3 (per strain) and n = 7 (per strain) at 56-days p.i. Statistical analysis (One-way ANOVA) was performed using Graphpad Prism software. Data represents mean and standard error mean.

## Genomic DNA extraction for WGS library preparation

Independent colonies of *Rv*, *RvΔung*, *RvΔudgB*, and *RvΔdKO* were selected randomly for whole genome sequencing, and no bias was given in terms of size or morphology while selecting colonies for sequencing. We picked independent colonies of *Rv* (n = 3), *RvΔung* (n = 4), *RvΔudgB* (n = 4) and *RvΔdKO* (n = 3) that were grown *in vitro*. Besides, we performed the whole genome sequencing of the colonies that were isolated from guinea pig lungs 56-days p.i. *Rv* (n = 11), *RvΔung* (n = 8), *RvΔudgB* (n = 8) and *RvΔdKO* (n = 9). We performed the whole genome sequencing of ciprofloxacin-resistant *RvΔdKO* (n = 13). *Rv*, *RvΔung*, *RvΔudgB*, and *RvΔdKO in vitro* cultures, or colonies obtained from guinea pig lung homogenate, or ciprofloxacin-resistant colonies, were grown till $A_{600}$ ~0.8 in 7H9-ADC medium (30 ml) and genomic DNA extracted as per manufacturer's instructions (Qiagen). The integrity of the isolated genomic DNA was analyzed by agarose gel electrophoresis, and the DNA was quantified using a Qubit fluorometer. Libraries were prepared using QIAseq FX DNA library Kit. Sequencing was performed using the Illumina Hiseq 4000 platform, generating paired-end reads of ~101 bp length. A total of 63 samples covering *Rv*, *RvΔung*, *RvΔudgB*, and *RvΔdKO* strains were sequenced. Fastq paired-end files were aligned on the H37Rv reference genome using Bowtie2 [27]. Generated SAM files were converted into BAM files using SAM Tools [28]. Qualimap was used to evaluate the alignment of the data as well as other quality points [29]. SNPs were extracted from BAM files using VarScan software [30]. Annotation of SNPs was performed using SnpEff toolbox [31]. All 63 VCF files were combined to create a matrix that includes

chromosome position, a nucleotide position in the reference genome, the identified SNP in the newly sequenced genomes, SNP biotype, amino acid change, and respective genes/intergenic region (S1, S3, S5, S7 and S9 Tables). For the identification of unique SNPs under different conditions, the laboratory strain of *Rv* was used as a reference.

## Ex vivo and in vivo competition experiments

Peritoneal macrophages were isolated from BALB/c mice 72 h post intraperitoneal thioglycollate injection. 5 x $10^5$ cells/well were seeded in the wells of a 24 well plate in RPMI medium. Cells were infected with *Rv*, *RvΔdKO*, and *RvΔdKO::ung-udgB* independently, or together at 1:1 ratio, at an M.O.I of 1:5. CFUs were enumerated 4 h p.i to determine the uptake. Cells were lysed at 36 h p.i in 0.05% SDS. Intracellular bacteria obtained after lysis were washed thrice using 1XPBS to remove SDS, and half the bacteria were used for CFU enumeration on 7H11-OADC plain plates while the remaining half was used for the next round of infection. The same procedure was followed for the third round of infection. *RvΔdKO* strain is hygromycin-resistant whereas *Rv* is sensitive to antibiotics. Colonies obtained on 7H11-OADC plates from mixed infection were patched on hygromycin- containing plates to score for *RvΔdKO* or *RvΔdKO::ung-udgB*. *Ex-vivo* infection was performed using two independent biological experiments and each biological experiment was performed in triplicates. Statistical analysis (Unpaired t-test) was performed using n = 3 for each biological experiment. Graphpad Prism software was employed for performing statistical analysis. Data represents one of the two biological experiments. Inbred BALB/c mice were challenged with *Rv* which has pST-Ki integrative plasmid that provides kanamycin resistance and *RvΔdKO* independently, or together at a 1:1 ratio. Briefly, 2 x $10^8$ cells of *Rv* or *RvΔdKO* or a 1:1 mix of both were suspended in 15 ml saline for performing infections through the aerosol route in mice. Similarly, 1 x $10^8$cells of *Rv* and *RvΔdKO* mix (1:1) or *Rv* and *RvΔdKO::ung-udgB* mix (1:1) were suspended in 15 ml saline for performing infections through the aerosol route in guinea pigs. CFUs were enumerated 1- and 56-days p.i., on 7H11-OADC with or without hygromycin/kanamycin. For *in-vivo* infection, statistical analysis was performed at day 1 using n = 3 (per strain) mice or guinea pigs and n = 8 mice or n = 7 guinea pigs (per strain) at 56-days p.i. Statistical analysis (Unpaired t-test) was performed using Graphpad Prism software. Data represents mean and standard deviation.

## MIC determination

*Rv*,*RvΔung*, *RvΔudgB*, and *RvΔdKO* strains were grown in 7H9-ADC medium to $A_{600}$~0.6 and diluted in fresh 7H9-ADC medium to obtain $A_{600}$~0.0006. Medium (100 μl of 7H9-ADC) was added to 96-well plate and ciprofloxacin was serially diluted. Cells (100 μl) were added to these wells and the plate was incubated for 5 days at 37˚C, followed by the addition of 20 μl of resazurin (0.2% w/v in AMQ) for 24 h.

## List of DNA oligomers

A list of oligonucleotides used in the study is provided in S11 Table. Source data file for all the figures is provided as S12 Table.

# Results

## Generation and characterization of uracil DNA glycosylases mutants in Mtb

To examine the impact of deletion of UDGs in the pathogen's survival and adaptability, we set out to delete *ung*, *udgB*, and both together in the drug-sensitive laboratory strain *H37Rv* (*Rv*).

The *ung* and *udgB* genes at the native loci were replaced with hygromycin resistant or chloramphenicol resistant cassette with the help of a specialized transduction method to generate *RvΔung* and *RvΔudgB*, respectively (Fig 1A and 1C). Recombination at the native loci was confirmed using multiple sets of PCR reactions (Fig 1B and 1D). Subsequently, to generate the combinatorial mutant, the native *udgB* gene in the *RvΔung* was replaced with chloramphenicol resistant cassette to generate *RvΔdKO* (Fig 1E). Western blot analysis using α-Ung and α-UdgB antibodies confirmed the deletion of *ung*, *udgB*, and both *ung-udgB* in the mutants (Fig 1F).

To ensure that the deletion of *ung* and *udgB* resulted in a loss of UDG activity, we performed biochemical assays. While Ung excises uracil residues from both the single-stranded and double-stranded contexts, UdgB removes uracil residues exclusively from the double-stranded substrates. We used both the single-stranded (SSU9) and double-stranded (GU9) DNA oligomers having uracil at the 9$^{th}$ position as substrates for monitoring the UDG activities (Fig 1G). Incubation of the 5'-$^{32}$P-radiolabeled substrate(s) with the cell-free extract would result in the formation of abasic site, which can be subsequently cleaved to generate $^{32}$P labeled smaller size product that migrates faster (Fig 1G and 1H). As expected, the radiolabeled band corresponding to the product could not be detected in *RvΔung* and *RvΔdKO* lysates with either SSU9 or GU9, suggesting the loss of UDG activity. Since the UDG activity of Ung is overwhelming compared with UdgB, to assess the activity of UdgB, it is necessary to inhibit the activity of Ung, which the addition of Ugi can achieve. Ugi is a small protein encoded by the *Bacillus subtilis* phage that specifically inhibits the activity of Ung at a 1:1 ratio [32]. Preincubation of lysates with Ugi abrogated the activity of Ung, which was apparent from the absence of the UDG activity on SSU9 (Fig 1H). Incubation of lysates from *Rv* and *RvΔung* with GU9 in the presence of Ugi resulted in the formation of the desired product indicating the activity of UdgB. However, no product formation was detected in *RvΔudgB* and *RvΔdKO* lysates suggesting successful deletion of *udgB* in these strains.

## Characterization of complementation strains of uracil DNA glycosylases

We next sought to investigate the growth kinetics of *RvΔung*, *RvΔudgB*, and *RvΔdKO* in nutrient-rich or limiting mediums. There were no discernible growth defects either in a rich or limiting medium under *in vitro* conditions suggesting that *ung* and *udgB* independently and together are dispensable for the *in vitro* growth (Fig 2A and 2B). Similar growth phenotypes were observed when the strains were subjected to various *in vitro* conditions of oxidative, nitrosative, and hypoxic stresses (S1A–S1D Fig). For the generation of complementation strains, *ung* and *udgB* genes were independently cloned into the integrative vector pST-Ki [24,33] under a constitutive promoter (P$_{smyc}$) to generate pST-Ki-*ung* and pST-Ki-*udgB* constructs, respectively. Next, we created a construct that expresses both *ung* and *udgB* by excising *udgB* and the promoter from pST-Ki-*udgB* and sub-cloning it into the unique SnaBI site pST-Ki-*ung* to generate pST-Ki-*ung-udgB* (Fig 2C). pST-Ki-*ung*, pST-Ki-*udgB* and pST-Ki-*ung-udgB* constructs were electroporated into *RvΔung*, *RvΔudgB*, and *RvΔdKO* strains to generate *RvΔung*::*ung*, *RvΔudgB*::*udgB*, and *RvΔdKO*::*ung-udgB*, respectively. UDG activity assays were performed to characterize the complementation strains. Complementation of *RvΔung* and *RvΔdKO* with pST-Ki expressing *ung* resulted in the restoration of UDG activity, which was apparent by the presence of radiolabelled product upon incubation of either SSU9 or GU9 with the lysates (Fig 2D). Similarly, we observed the complementation of UdgB activity in the presence of Ugi when lysates from *RvΔudgB*::*udgB*, and *RvΔdKO*::*ung-udgB* were used (Fig 2E). Collectively, these data confirm the generation of the desired gene deletion mutants and the corresponding complementation strains.

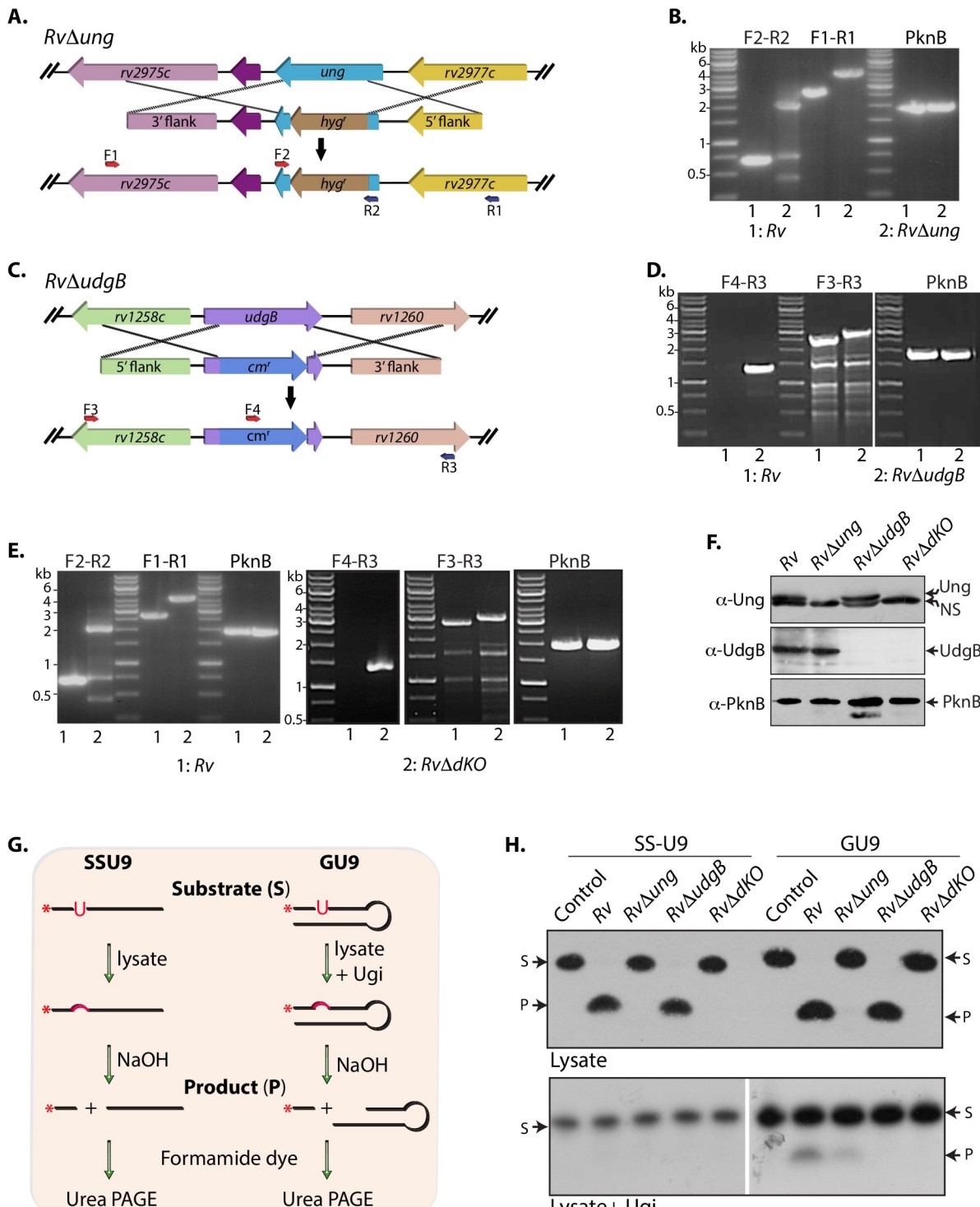

**Fig 1. Generation and characterization of uracil excision repair mutants. (A)** Schematic depicting the generation of gene replacement mutant of *ung*. The insertion of hygromycin resistance cassette disrupted the native allele. **(B)** PCR using F2-R2 (gene-specific primers) resulted in the amplification of 684 bp in *Rv* and ~2 kb in *RvΔung*. PCR using F1-R1 (primers beyond the 5' and 3' flank) resulted in the amplification of ~3 kb in *Rv* and ~4.5 kb in *RvΔung*. PknB gene amplification was used as the positive control. **(C)** Schematic depicting the generation of gene replacement mutant of *udgB*. The insertion of chloramphenicol resistance cassette disrupted the native allele. **(D)** PCR using F4-R3 resulted in amplification in *RvΔudgB* but not in *Rv*. PCR using F3-R3 resulted in amplification of ~2.5 kb and ~3 kb in *Rv* and *RvΔudgB*, respectively. **(E)** In the background of *RvΔung*, the *udgB* native allele was disrupted by the insertion of the chloramphenicol resistance cassette. Indicated primers were used for screening the *RvΔdKO*. **(F)** Immunoblot analysis for the confirmation of gene

replacement mutants. 50 and 100 µg of *Rv*, *RvΔung*, *RvΔudgB*, and *RvΔdKO*. WCLs were resolved on 16% Tris-Tricine-Urea gel, transferred to a nitrocellulose membrane, and probed with α-Ung (1:2000) and α-UdgB (1:2000) antibodies, respectively. WCL (10 µg) was resolved on 10% SDS-PAGE, transferred to a membrane, and probed with α-PknB (1:10000) antibody. (G) Schematic representation of UDG activity assay. SSU9 and GU9 were incubated with various lysates or Ugi. (H) Cell extracts (10 µg) of *Rv*, *RvΔung*, *RvΔudgB*, and *RvΔdKO* and the 5'-$^{32}$P end-labeled SSU9 and GU9 (25000 c.p.m.) were used for performing UDG assay. Ugi (25ng) was preincubated with *Rv*, *RvΔung*, *RvΔudgB*, and *RvΔdKO* for performing UdgB activity assays. Product and Substrate are labeled as 'P' and 'S'.

## Uracil DNA glycosylases mutants exhibit hypermutability

Since *ung* and *udgB* encode the key base excision repair enzymes, we sought to determine the accumulation of mutations in the genomes of *RvΔung*, *RvΔudgB*, and *RvΔdKO* under the normal laboratory growth conditions (Fig 3A) by performing whole-genome sequencing (WGS). Analysis of the data revealed that the laboratory *Rv* strain showed 92 SNPs, which include synonymous, non-synonymous, non-coding, and intergenic, compared with the reference *H37Rv* genome (S1 and S2 Tables). The majority of the SNPs were found in PPE and PE_PGRS family genes (S2A Fig) and we observed all possible transition and transversion mutations (S2B Fig). PE and PPE family of proteins in *Mtb* represent ~10% of its coding regions. Due to the presence of repetitive sequences and high GC rich content, these regions cannot be sequenced and aligned with high confidence [34], which may contribute to a higher number of SNPs observed in the laboratory *Rv* strain compared with the reference genome. *RvΔung*, *RvΔudgB*, and *RvΔdKO* were generated in laboratory *Rv* background (*Rv in vitro*); therefore, in subsequent analysis, we have used *Rv in vitro* as the reference genome. Next, we analyzed the unique SNPs found in *RvΔung*, *RvΔudgB*, and *RvΔdKO* compared with *Rv in vitro* (Fig 3B and S2C and S3 and S4 Tables). Interestingly, we observed very few SNPs in *RvΔung*, *RvΔudgB*, and *RvΔdKO* strains; most of them were also found in the PPE and PGRS genes. The observed SNPs did not show specific mutation spectrum such C➜T or G➜A mutations, suggesting that the deletion of UDGs did not result in accumulation of significant mutations when cultured in a nonselective complete medium (S2C Fig).

Acquisition of spontaneous mutations under stress conditions can either lead to cell death if the mutations are deleterious or improved survival if the mutations are advantageous [35]. To examine the acquisition of spontaneous mutations in the absence of *ung*, *udgB*, or both, we performed mutation rate analysis using a fluctuation test in the presence of rifampicin and isoniazid (Figs 3C and 3D and S2D)[25]. *RvΔung*, *RvΔudgB*, and *RvΔdKO* strains' spontaneous mutation rate was significantly higher than *Rv* (Fig 3D and 3E). Deletion of *ung*, *udgB*, or both resulted in a 3.57, 4.3, and 22.28-fold increase in the spontaneous mutation rate in the presence of rifampicin, respectively. While the phenotype was completely restored in *RvΔudgB*::*ungB* and *RvΔdKO*::*ung-udgB*; *ung* complementation resulted only in partial restoration. Similarly, we observed a 5.32, 22.11, and 37.17-fold increase in the spontaneous mutation rate of *RvΔung*, *RvΔudgB*, and *RvΔdKO* strains compared with *Rv* in the presence of isoniazid (Fig 3D and 3E). The phenotype was restored in the complementation strains. Collectively the data suggest that deletion strains exhibit a higher spontaneous mutation rate (Figs 3D and 3E and S2E). The Rifampicin Resistance Determining Region (RRDR) sequence of the spontaneous rifampicin resistant colonies revealed multiple mutations in the loci (S3B Fig). In Rv, most of the mutations were in RRDR, in *RvΔung*, *RvΔudgB*, we also found mutations outside RRDR. Interestingly, in *RvΔdKO*, we found mutations in Ile-488 outside the RRDR region, a residue suggested to be involved in interactions with rifampicin [36].

Bacteria encounter reactive oxygen species (ROS) and reactive nitrogen intermediates that cause DNA damage inside macrophages [37]. Therefore, we examined the mutation frequencies in the presence of acidified sodium nitrite (source of RNI) and cumene hydroperoxide

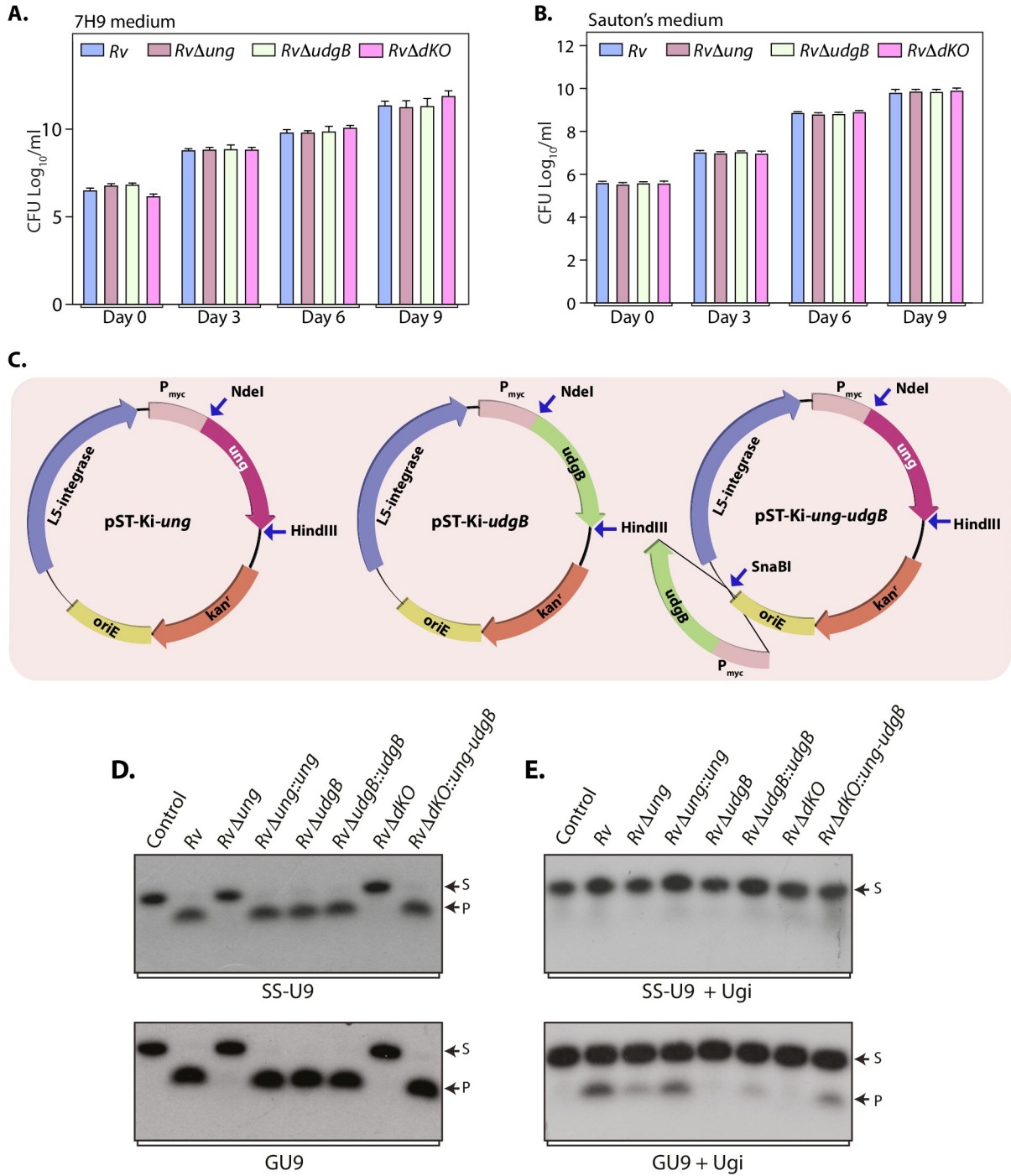

**Fig 2. Characterization of complementation strains of uracil DNA glycosylases. (A and B)** Growth profiles of *Rv*, *RvΔung*, *RvΔudgB*, and *RvΔdKO* in 7H9 or Sauton's minimal medium were determined by CFU enumeration on 7H11-OADC plates at indicated time points. Data represent two biologically independent experiments. Each experiment was performed in quadruplets. Data represent mean and standard deviation. **(C)** Schematic representation of complementation constructs pST-Ki-*ung*, pST-Ki-*udgB*, pST-Ki-*ung-udgB*. **(D and E)** UDG activity assays were performed in the absence or presence of Ugi using *Rv*, *RvΔung*, *RvΔudgB*, and *RvΔdKO*, *RvΔung::ung*, *RvΔudgB::udgB* and *RvΔdKO::ung-udgB* as described in Methods.

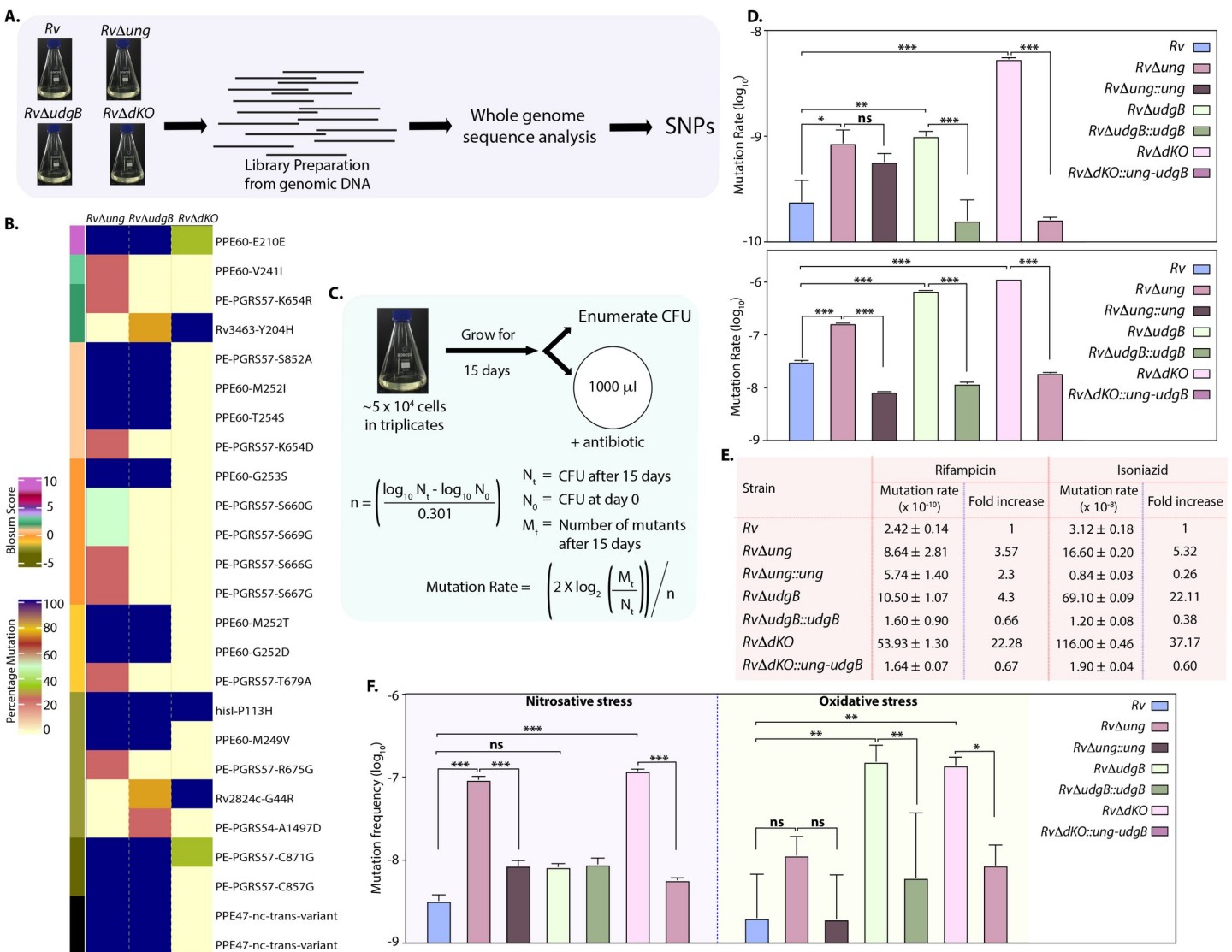

**Fig 3. Uracil DNA glycosylase gene mutants exhibit hypermutability. (A)** Mid-log-phase cultures of *Rv*, *RvΔung*, *RvΔudgB*, and *RvΔdKO* were used for genomic DNA preparation. Schematic showing the procedure employed for WGS. **(B)** WGS of *RvΔung*, *RvΔudgB*, and *RvΔdKO* were compared with WGS of *Rv* grown *in vitro*. The heat map shows the percent SNPs present in *RvΔung* (n = 4), *RvΔudgB* (n = 4), and *RvΔdKO* (n = 3). Blosum score provides information about the synonymous and non-synonymous changes due to SNP. **(C)** Schematic representation of spontaneous mutation rate analysis. The mutation rate was calculated using a fluctuation test. **(D)** Rifampicin and isoniazid resistance rates for *Rv*, *RvΔung*, *RvΔung::ung*, *RvΔudgB*, *RvΔudgB::udgB*, *RvΔdKO* and *RvΔdKO::ung-udgB*. Data is representative of biological triplicates. Graphpad prism was used for the statistical analysis (one-way ANOVA). Data represent mean and SD. **(E)**The table represents a fold increase of the mutation rate of *Rv*, *RvΔung*, *RvΔung::ung*, *RvΔudgB*, *RvΔudgB::udgB*, *RvΔdKO* and *RvΔdKO::ung-udgB* with respect to *Rv*. **(F)** Mutation frequency was calculated after subjecting the *Rv*, *RvΔung*, *RvΔung::ung*, *RvΔudgB*, *RvΔudgB::udgB*, *RvΔdKO* and *RvΔdKO::ung-udgB* to nitrosative or oxidative stress, respectively. Cells were plated on no antibiotic and rifampicin (10 μg/ml)-containing plates. Data represent one of the two biological experiments and each experiment was performed in triplicates. Data represent mean and SD. *p<0.01, **p<0.001 and *** p<0.0001.

(CHP) (source of ROS). In the presence of acidified sodium nitrite *RvΔung* and *RvΔdKO* showed higher mutation frequency than *Rv* and *RvΔudgB;* most likely due to increased deamination of cytosines, which would accumulate in the absence of *ung* (Fig 3F). On the other hand, in the presence of CHP, mutation frequency observed with *RvΔudgB* and *RvΔdKO* were higher compared with *Rv* and *RvΔung*, which is likely because, besides uracils, UdgB is known to excise hypoxanthine and ethenocysteine from DNA (Fig 3F). The complementation strains

restored the phenotype in both the stress conditions. These results suggest the differential abilities of Ung and UdgB in repairing DNA damage in the host.

## Uracil DNA glycosylase mutants exhibit hypervirulent phenotype

*In vitro* data suggested that the UDG mutants have a higher spontaneous mutation rate (Fig 3D and 3E), thus we sought to evaluate the stress induced mutagenesis of all the strains under hostile *in vivo* conditions. Towards this we investigated the impact of deleting UDGs on the survival of the pathogen in the host using a guinea pig infection model. We first assessed the survival of *Rv*, *RvΔung*, *RvΔudgB*, and *RvΔdKO* in guinea pigs (S4A Fig). Implantation of *Rv*, *RvΔung*, *RvΔudgB*, and *RvΔdKO* at day 1 p.i. was comparable (S4B Fig). CFU analysis at 56-days p.i. did not result in a significant increase in *RvΔung* and *RvΔudgB*'s survival compared with *Rv*, whereas *RvΔdKO* showed a ~0.5 log fold increase (S4B Fig). Compared with the *in vitro* grown *Rv* strain, WGS analysis of *Rv* colonies obtained from infected guinea pig lungs showed minimal SNPs accumulation. As was the case with *in vitro* growth, these occurrences did not show any specific bias in the mutation spectrum (Fig 4A–4C and S5 and S6 Tables). On the other hand, WGS analysis of the colonies recovered p.i. showed significant SNP accumulation in UDG mutant strains. Most importantly, colonies of *RvΔudgB* and *RvΔdKO* showed a much higher number of SNPs compared with *RvΔung*. The SNPs were mostly in the genes encoding proteins involved in intermediary metabolism, cellular respiration, cell wall, cell processes, conserved hypotheticals, PE/PPE family, and lipid metabolism (Figs 4D and S4C, and S7 Table). Since *RvΔdKO* exhibit a better survival in the guinea pig lungs and showed a higher accumulation of SNPs, therefore, we sorted the observed SNPs in *RvΔdKO* based on the nature of mutation (Fig 4F). The analysis showed that 34% of mutations are synonymous, 58% mutations are non-synonymous, 6.4% are intergenic mutations, and 1.6% have non-sense mutations.

Further analysis showed that genes such as *vapC47*, *gadB*, *eccD4*, *PE-PGRS9*, and *fadD4* possess non-synonymous mutations. Deletion of any of these genes provide a growth advantage to the pathogen [38]. Interestingly, a non-sense mutation in the *Rv3437* at Glu-69 position was identified in *RvΔdKO*. Disruption of this gene also provides a growth advantage to the bacteria (https://mycobrowser.epfl.ch/) (S4D Fig). However, we did not identify mutations in the drug resistance-conferring loci such as *inhA*, *rpoB*, *or katG*. This is most likely because guinea pigs were not subjected to antibiotics treatment before or after the infection. Spectrum analysis of mutations showed a clear predisposition towards G→A and C→T mutations, which is most likely due to these strains' inability to repair modified cytosines and a few other bases (Fig 4E and S8 Table). The mutation spectrum analysis was performed by counting the number of independent mutations per gene; therefore, C→T mutation is different from G→A mutation. These results suggest that the UDG mutants show a superior ability to evolve and survive under *in vivo* stresses enforced by the host.

There exists a possibility that the hypervirulent phenotype observed in *RvΔdKO* (S4B Fig) could be due to the spontaneous reversion of attenuating mutations. To negate this possibility, we reperformed the guinea pig infection experiment with *Rv*, *RvΔdKO*, and the complementation strain *RvΔdKO::ung-udgB*. Implantation of *Rv*, *RvΔdKO* and *RvΔdKO::ung-udgB* at day 1 p.i in guinea pigs' lungs was comparable (Fig 4G and 4H). While the CFUs obtained 56-days p. i were similar for *Rv*, and *RvΔdKO::ung-udgB*, we noticed an ~0.5 log fold increase in the CFUs of *RvΔdKO* in lungs and spleen (Fig 4H). Hematoxylin and eosin staining performed to examine the gross histopathology showed the presence of well-formed granulomas in all three strains (Fig 4I). The results are consistent with the earlier guinea pig infection experiment (S4B Fig). The ability of complementation strain to restore the phenotype suggested that *RvΔdKO's* hypervirulence is not due to the spontaneous reversion of attenuating mutations.

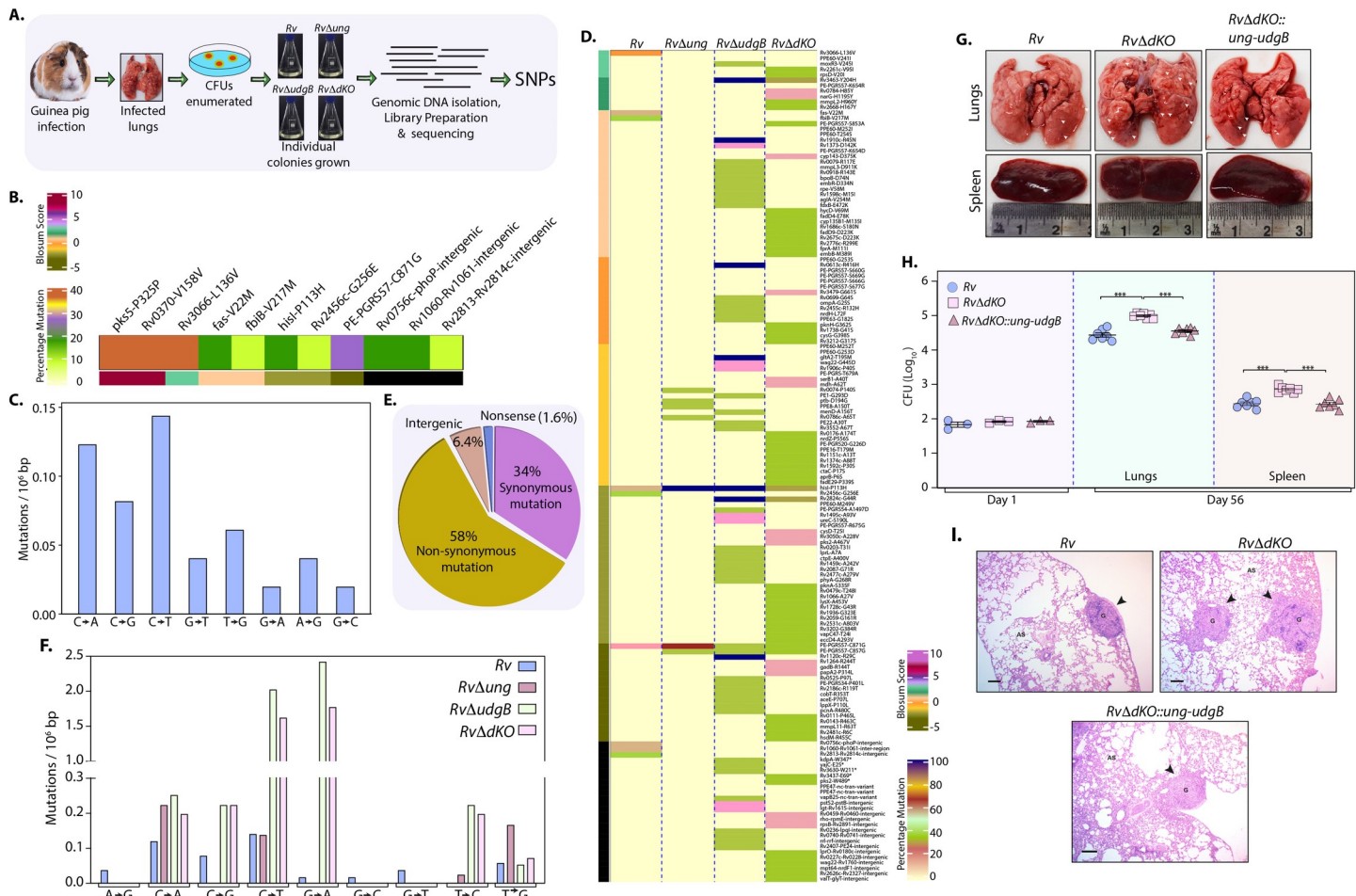

**Fig 4. Uracil DNA glycosylase mutants exhibit hypervirulent phenotype. (A)** Schematic outline depicting the pipeline used for WGS of colonies obtained from the guinea pig lungs *Rv* (n = 11), *RvΔung* (n = 8), *RvΔudgB* (n = 8), and *RvΔdKO* (n = 9). **(B)** *Rv* sequenced from guinea pig lungs was compared with the sequence of *Rv* grown *in vitro*. A heat map represents the unique SNPs identified in the *Rv* isolated from guinea pig lungs. Blosum score provides information about the synonymous and non-synonymous changes due to SNP. **(C)** SNPs per million nucleotides in the sequenced *Rv* genome, isolated from guinea pig lungs. **(D)** Heat map showing the SNPs accumulated in *Rv*, *RvΔung*, *RvΔudgB*, *RvΔdKO* isolated from guinea pig lungs compared with the sequence of *Rv* grown *in vitro*. Blosum score provides information about the non-synonymous changes due to SNP. Percentage mutation provides information about the percent of strains sequenced where a particular SNP is detected **(E)** Nature of mutations identified in the RvΔdKO isolated from guinea pigs. **(F)** SNPs per million nucleotides in the sequenced *Rv*, *RvΔung*, *RvΔudgB*, and *RvΔdKO*. Mutations listed are for one strand of DNA. The mutation spectrum analysis was performed by counting the number of independent mutations per gene; therefore, C➜T mutation is different from G➜A mutation E **(G-I)** Guinea pigs were challenged with *Rv*, *RvΔdKO*, and *RvΔdKO::ung-udgB* strains. At 56-days p.i., lungs and spleen were isolated. Gross histopathology examined by hematoxylin and eosin staining. CFUs were enumerated by plating lung and spleen homogenate for all seven. **(G)** Representative gross pathology images of the infected guinea pig lungs and spleen. White arrows shows the discrete tubercle. **(H)** CFUs were enumerated as described in Methods. Data represent CFU (log$_{10}$) per lung at day 1 p.i. CFU (log$_{10}$/ml) from the lungs and spleen of infected guinea pigs (n = 7/per strain) at 56-days p.i. Statistical analysis (one-way ANOVA) was performed using Graphpad prism. ***$p<0.0001$. **(I)** Gross histopathology images (40x- magnification) of *Rv*, *RvΔdKO*, and *RvΔdKO::ung-udgB* of infected guinea pig lungs. Black arrows indicate granuloma. **G**-Granuloma, **AS**-Alveolar Space.

## The uracil DNA glycosylase mutants accelerate the acquisition of antibiotic resistance

The data show that the *RvΔdKO* exhibits a survival advantage over the wild-type parent under drug selection conditions or the host-imposed stress. Mutations in UDGs could help in the faster evolution of drug resistance. If this were indeed the case, *RvΔdKO* could also be an excellent resource to ascertain mutations at the genome-scale that could result in resistance against a candidate drug. To check the prediction, as a test, we set out to identify mutations that would make the strain resistant to second-line antibiotic ciprofloxacin. Ciprofloxacin MIC values

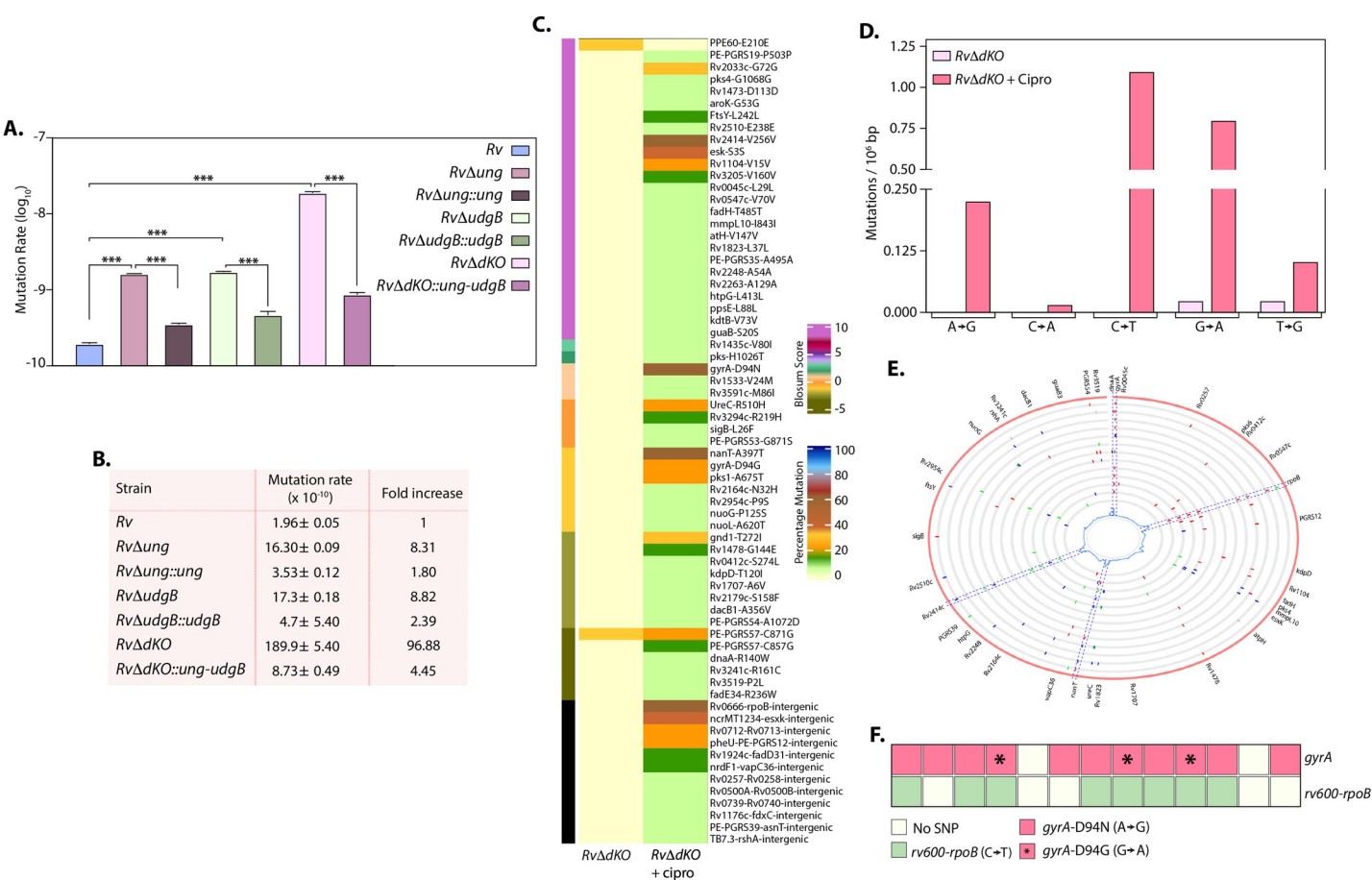

**Fig 5. Uracil DNA glycosylase mutants accelerate the acquisition of antibiotic resistance. (A)** Spontaneous mutation rate analysis was performed using ciprofloxacin. Data are representative of one of the two biological experiments. Each biological experiment was performed in triplicates. Data represent mean and SD. Statistical analysis (one way ANOVA) was performed using Graphpad prism. ***$p<0.0001$. **(B)** Table shows ciprofloxacin resistance rate calculated as described in Fig 2C for *Rv*, *RvΔung*, *RvΔung::ung*, *RvΔudgB*, *RvΔudgB::udgB*, *RvΔdKO* and *RvΔdKO::ung-udgB*. Fold increase of the mutation rate of *Rv*, *RvΔung*, *RvΔung::ung*, *RvΔudgB*, *RvΔudgB:: udgB*, *RvΔdKO*, and *RvΔdKO::ung-udgB* with respect to *Rv*. **(C)** Heat map showing the SNPs accumulated in *RvΔdKO* (n = 3) grown *in vitro* and ciprofloxacin-resistant *RvΔdKO* (n = 13) with respect to *Rv* grown *in vitro*. Blosum score provides information about the synonymous and non-synonymous changes due to SNP. Percentage mutation provides information about the percent of strains sequenced where a particular SNP is detected. **(D)** SNPs per million nucleotides in the sequenced *RvΔdKO* grown *in vitro* and ciprofloxacin-resistant *RvΔdKO*. **(E)** Circos plot representing the mutations in the genome of ciprofloxacin-resistant *RvΔdKO* (grey circles; n = 13) with respect to *Rv* reference sequence (red circle). The spikes in the innermost circle show mutation frequencies. Mutations in *gyrA*, *rv2414c*, *nanT*, and *rpoB* intergenic regions are highlighted with dotted blue lines. **(F)** A matrix representing mutation in the *gyrA* and *Rv600-rpoB* intergenic region in ciprofloxacin-resistant *RvΔdKO*.

obtained for *Rv*, *RvΔung*, *RvΔudgB*, and *RvΔdKO* were comparable (MIC- 0.31 µg/ml), suggesting that at the outset both the strains were sensitive to the antibiotic (S4E Fig). The fluctuation test revealed that the mutation rate was 8.31, 8.82 and 96.88-fold higher in *RvΔung*, *RvΔudgB*, and *RvΔdKO*, respectively compared with the *Rv* strain (Fig 5A and 5B), and the complementation of *ung* and *udgB* independently or together rescued the phenotype (Fig 5A and 5B). WGS of 13 independent ciprofloxacin-resistant colonies showed that *RvΔdKO* acquired many mutations compared with the antibiotic-sensitive naïve *RvΔdKO* parent (Fig 5C and S9 Table). As anticipated, most of the mutations were G➡A or C➡T (Fig 5D and S10 Table). Circos plot constructed using *Rv* as the reference genome showed synonymous, non-synonymous, and intergenic mutations (Fig 5E).

Additionally, a mutation was also present in the intergenic region of *Rv600-rpoB* (Fig 5F). However, the biological impact of the identified mutation is unclear. Importantly, the analysis

showed mutations in *gyrA*, *nanT*, and *ureC*, in multiple ciprofloxacin-resistant strains ($\geq$40%). In addition to mutations in *gyrA*, a direct target of fluoroquinolones, we identified novel mutations; an example is mutation in *ureC* that encodes for urea amidohydrolase. Also we found mutations in *nanT*, which encodes an integral membrane protein involved in sialic acid transport. A➔ G or G➔A mutation in *gyrA* that results in Asp➔Asn or Gly at the 94th position occur in all fluoroquinolone-resistant isolates [39]. Together, the results suggest that the absence of UDGs results in faster evolution of antibiotic resistance by the acquisition of mutations and therefore can be an excellent tool for identifying targets responsible for drug resistance.

## *RvΔdKO* strain displays superior fitness *ex vivo* and *in vivo*

The fitness of a bacterial strain derives from its ability to survive better than the other bacteria in a given environment. Independent survival of both *Rv* and *RvΔdKO* was comparable in the peritoneal macrophages (pΦ), RAW, and THP1 cells (S5A–S5C Fig). Subsequently, to evaluate the strains' relative fitness, we infected pΦ with *Rv*, *RvΔdKO* and, *RvΔdKO*::*ung-udgB* independently or in combination wherein we competed *Rv* with *RvΔdKO* or *Rv* with *RvΔdKO*::*ung-udgB*. We reasoned that the differences between *Rv* and *RvΔdKO* or *Rv* with *RvΔdKO*::*ung-udgB* strains would only be apparent if we continuously compete strains against each other by performing consecutive rounds of infections with the bacterial population recovered from the preceding infections (Fig 6A). Thus, bacilli obtained 36 h after the first round of infection were used to infect fresh pΦ cells and the whole process was repeated twice over (Fig 6A). When the experiment was performed with *Rv*, *RvΔdKO*, and *RvΔdKO*::*ung udgB* strains independently, their proficiency in all the sets was comparable (Fig 6B). The total bacillary survival in *Rv* + *RvΔdKO* or *Rv* + *RvΔdKO*::*ung-udgB* competition experiment was also comparable with independent infection (Fig 6B). However, when the strains were rivaled against each other, *RvΔdKO* outcompeted *Rv* with each subsequent round of infection (Fig 6C). While the survival abilities were comparable after the first round of infection (Fig 6C; 2nd set), after completing second and third rounds of infection, *RvΔdKO* showed its clear dominance over *Rv* (Fig 6C; 3rd and final set). On the other hand, competition between *Rv* and *RvΔdKO*::*ung-udgB* did not show discernable differences in the survival, suggesting; a. expression of *ung* and *udgB* from the L5 *att* site rescued the phenotype; b. the survival advantage is indeed due to the absence of UDGs.

*In vivo*, necrosis of infected cells and subsequent spread of infection to the neighboring cells is responsible for the increased bacillary load. If a strain has superior fitness over the other, one would expect that it would show relatively higher CFUs with time. Even though, independently *Rv* and *RvΔdKO* strains showed comparable CFUs 56 days p.i., when competed against each other *RvΔdKO* exhibited a decisive advantage over *Rv* (Fig 6D). We next performed a competition experiment between *Rv* and *RvΔdKO* or *Rv* and *RvΔdKO*::*ung-udgB* using the guinea pigs model of infection. CFU plating at 56 days p.i suggests ~0.5 to 1 log fold increase in the survival of *RvΔdKO* in comparison with *Rv* in lungs and spleen, respectively (Fig 6E). In case of *Rv* and *RvΔdKO*::*ung-udgB*, no apparent difference was observed (Fig 6E). Percent CFUs analysis showed an equal deposition of strains at day 1 in mice and in guinea pigs lungs. Survival difference between *Rv* and *RvΔdKO* became evident at day 56 p.i in the lungs and spleen (Figs 6E and 7A). Together, these results suggest that *RvΔdKO* displays a decisive edge over the wild-type parent (Fig 7B). We speculate that this is due to the strain's ability to develop mutations that provide a survival advantage. Collectively, deletion of BER genes *ung* and *udgB* drives the accumulation of mutations in the genome under stress conditions and assists in enhanced adaptation.

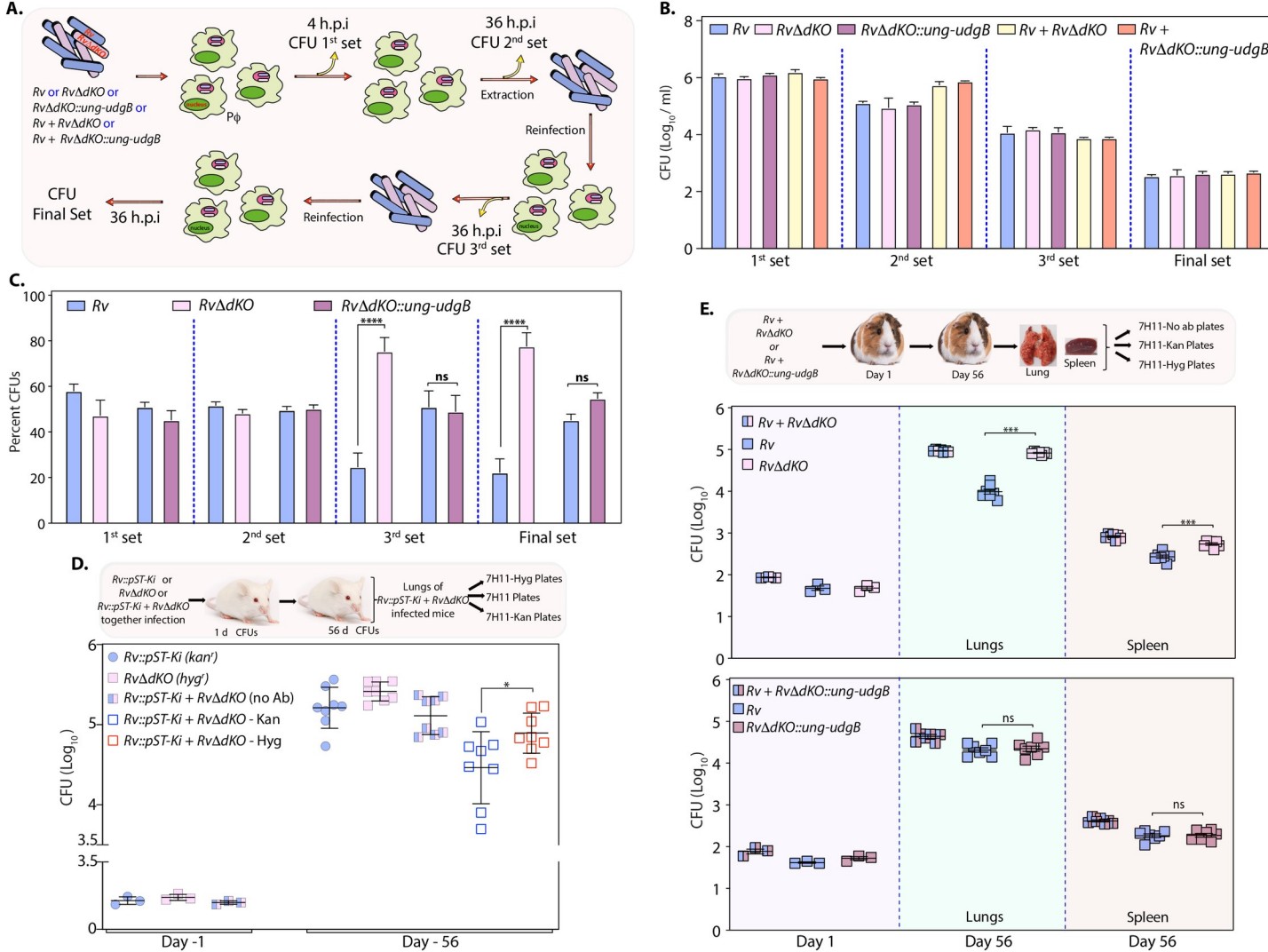

**Fig 6. RvΔdKO strain displays superior fitness ex vivo and in vivo. (A)** Schematic outline of the competition experiment performed in peritoneal macrophages. **(B)** Single-cell suspensions of *Rv*, *RvΔdKO*, and *RvΔdKO::ung-udgB* were used to infect peritoneal macrophages. Cells were lysed at the indicated time point, and CFUs were enumerated on 7H11-agar plates. Survival of *Rv*, *RvΔdKO*, and *RvΔdKO::ung udgB* independently at indicated time point shown in **(B)**. **(C)** Percent CFUs of *Rv* and *RvΔdKO* and *RvΔdKO::ung udgB* were determined by replica plating 100 colonies 7H11-hygromycin plates. While *RvΔdKO* or *RvΔdKO::ung-udgB* colonies can grow on 7H11-hygromycin plates, colonies belonging to *Rv* would not grow on these plates. *Ex-vivo* infection was performed using two independent biological experiments, and each biological experiment was performed in triplicates. Statistical analysis (Unpaired t-test) was performed using n = 3 for each biological experiment. Graphpad Prism software was employed for performing statistical analysis. *** $p < 0.0005$. Data represents one of the two biological experiments. **(D)** Schematic representation of a competition experiment in mice (n = 8). *Rv* (kanamycin-resistant) and *RvΔdKO* (hygromycin resistant) independently or together were challenged through the aerosol route. CFUs were enumerated at day 1- and 56-days p.i. on 7H11 plates as indicated. **(E)** Schematic representation of a competition experiment in guinea pigs (n = 7). *Rv* and *RvΔdKO* or *Rv* and *RvΔdKO::ung-udgB* together were challenged through the aerosol route. CFUs were enumerated at day 1- and 56-days p.i. on 7H11 plates as indicated. CFU(log$_{10}$) per lung was plotted at day 1 p.i. CFU (log$_{10}$/ml) was plotted at 56-days p.i. The graph shows CFUs representing the survival of *Rv* and *RvΔdKO* or *Rv* and *RvΔdKO::ung-udgB*. For *in vivo* infection experiment, statistical analysis was performed at day 1 using n = 3 (per strain) mice or guinea pigs and n = 8 mice or n = 7 guinea pigs (per strain) at 56-days p.i. Statistical analysis (Unpaired t-test) was performed using Graphpad Prism software. Data represents mean and standard deviation. *** $p < 0.0005$.

## Discussion

*Mtb* encodes for multiple DNA repair pathways such as base excision repair (BER), nucleotide excision repair (NER), homologous recombination (HR), Non-Homologous End Joining (NHEJ), and Single Strand Annealing (SSA) pathways. These pathways are crucial for maintaining the genomic integrity [40]. Mutations in DNA repair genes and their roles in the

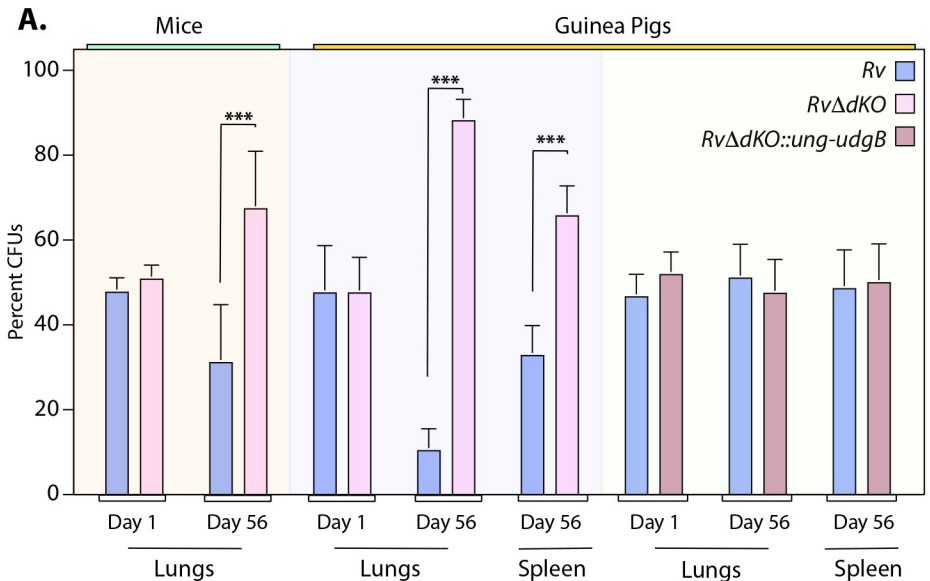

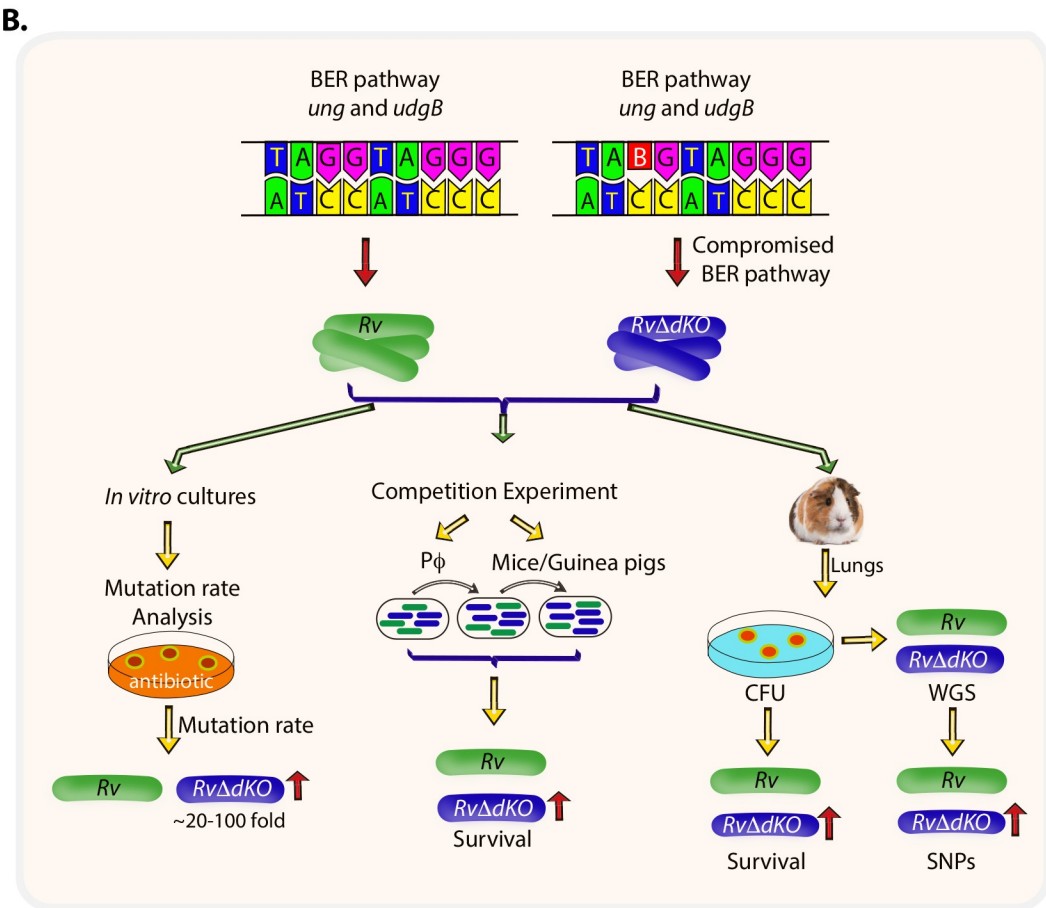

**Fig 7. *RvΔdKO* displays a decisive edge over the parent *Rv* strain. (A)** Graph represents percent CFUs calculated at day 1 and 56-days p.i. in the mice lungs and guinea pigs' lungs or spleen. Statistical analysis (Unpaired t-test) was performed at day 1 using n = 3 (per strain) mice or guinea pigs and n = 8 mice or n = 7 guinea pigs (per strain) at 56-days p.i. Data represent mean and standard deviation.*** *p*<0.0005. **(B)** The model depicts the biological implication of deletion of uracil DNA glycosylases. *In vitro*, *ex vivo*, and *in vivo* experiments suggest that deletion of uracil DNA glycosylases provides a survival advantage in the host.

emergence of antibiotic resistance were previously reported in various pathogens such as *Pseudomonas aeruginosa*, *Helicobacter pylori*, *Neisseria meningitides*, and *Salmonella typhimurium*. Among the *P. aeruginosa* strains isolated from cystic fibrosis (CF) patients, 36% had deletion or point mutations in DNA repair genes, *mutS*, *mutL*, and *mutY* [41]. Within *H. pylori* isolates from dyspeptic patients, 25% of strains showed high mutation frequency due to altered DNA repair and/or proofreading [42]. Similarly, 11% of epidemic strains of *N. meningitidis* possess defects in mismatch repair pathway genes *mutL or mutS* [43]. Recently, Mfd, a transcription-coupled NER factor, was implicated in the evolution of antibiotic resistance in *Bacillus subtilis*, *S. typhimurium*, and *P. aeruginosa* [44]. These findings establish a role for DNA repair genes in imparting drug resistance. Thus, we set out to investigate how the deletion of the DNA repair pathway in *Mtb* affects the survival in the host. We also sought to determine the contribution of the host environment in the acquisition of mutations.

Uracil residues in DNA arise due to the deamination of cytosine residues, which in the absence of repair results in CG➔TA mutations in the genome. Deletion of *ung* and *udgB* in *M. smegmatis* compromised bacterial survival under *in vitro* stresses [21]. However, in *Mtb*, the *ung* and *udgB* deletion strains did not show any survival defects in either *in vitro* or *in vivo* (S1 and S5 Figs). These results are in line with previously published work where *Mtb* genes involved in NHEJ, HR, and BER pathways were found to be dispensable for *in vitro* growth as well as *in vivo* survival. Deletions of genes of the NHEJ and HR double-stranded DNA repair pathways independently or together did not compromise *Mtb* survival in different animal infection models [45]. Deletion of AP-endonucleases, *endA*, and *xthA*, independently or together had no impact on mycobacterial survival in a guinea pig model of infection [46]. In the current study, while the survival of *ung* (*RvΔung*) and *udgB* (*RvΔudgB*) mutants individually was comparable to the wild type parent, double mutants of *ung* and *udgB* (*RvΔdKO*) showed improved survival in the guinea pig model of infection (Figs 4H and S4B). WGS analysis of the wild type and mutants grown under laboratory conditions did not show significant SNPs accumulation (Fig 3B). Suggesting that in the absence of external stress, the presence or absence of a particular repair gene may be of little consequence as the alternate DNA repair pathways may substitute for the deficiency. However, strains subjected to nitrosative or oxidative, or antibiotic stress, repair mutants displayed hypermutability (Figs 3D–3F). In line with this, WGS analysis of the wild type and mutant bacteria from the guinea pig's lungs showed a significant accumulation of mutations in the mutants compared with the wild type (Fig 4D). Non-synonymous or non-sense mutations in the genes that confer growth advantage to the bacteria upon deletion were identified in the *RvΔdKO* suggesting that the better survival of the is attributed to the accumulation and selection of advantageous mutations (S4D Fig). The difference was apparent when we compared *RvΔdKO* with *Rv*. DNA repair genes such as UDGs or other BER pathway genes, upon deletion, exhibit approximately 20% higher mutation rate, which would help in bacillary survival [47,48]. How is such a property advantageous to the pathogen under physiological conditions? Many DNA repair enzymes, including *ung* and *udgB* are downregulated during hypoxia [49]. This may explain the broad mutation spectrum observed in the strains isolated from the infected guinea pig lung granulomas, an oxygen depleted environment (Fig 6F). An advantage of such a regulation could be that when *Mtb* reactivates from dormancy, it carries a repertoire of population that offers better fitness to the bacterium to sustain itself for active growth in the changed physiological environment. The bacterium that reactivate from dormancy may also offer potential advantages in evolving/acquiring drug resistance. In contrast, deletion of genes such as dnaE1 or endoMS/nucS results in a very high mutation rate that eventually compromises the strain's fitness [50,51].

A classical way to isolate drug-resistant mutants is to select spontaneous resistant mutants *in vitro* by growing the laboratory strains in drug-containing media. Subsequent WGS analysis

of bacteria of the resistant colonies yields information on the loci responsible for resistance. Such methods have been used to identify multiple drugs' mechanism, including bedaquiline and pyrazinamide [34]. However, selecting for spontaneous resistant mutants can be time-consuming and at times may not be effective. Mutants of DNA repair gene intrinsically show higher mutation rates. Thus, we speculated that the double mutant of DNA repair genes (*RvΔdKO*) would be an excellent strain to expedite drug resistance selection. In line with this, we observed an ~97-fold higher mutation rate when selecting for resistance against the second-line drug ciprofloxacin (Fig 5A–5F). Most importantly, 11 out of the 13 resistant mutants sequenced showed mutations in *gyrA*, a well-known target of fluoroquinolones. Interestingly, in addition to mutations in the drug's direct target, we have identified mutations in *nanT*, *ureC*, *rv2414c* and intergenic region of *rv600-rpoB*. Further investigations are necessary to determine the role of these novel mutations in conferring fluoroquinolone resistance. Our results demonstrate that mutants of DNA repair genes can be successfully employed for faster acquisition of drug resistance, which can be employed in studies targeting the identification of mechanisms of drug resistance.

Fitness in evolutionary theory is measured using the competition assays wherein two strains compete for the same niche, which eventually results in the selection of the strain that has higher capabilities for adaptation [52]. Results show that *RvΔdKO* when competed against *Rv*, displayed enhanced proficiency in its ability to survive both *in vitro* and *in vivo*, suggesting that the mutant strain's fitness is superior to that of the wild-type strain (Fig 6). Our data suggest that bacteria harboring SNPs that are not favorable for survival are eliminated. Only those that acquire SNPs that confer a survival advantage are retained. We conclude that under duress, either due to host-induced stresses or antibiotic treatment, the *RvΔdKO* mutant can accumulate mutations that would accelerate the process of its natural selection. While DNA repair genes such as UDG are critical for maintaining genome integrity, their compromised function may help in the accumulation of mutations that provide survival advantage under unfavorable conditions. Collectively, data suggests that the deletion of UDGs results in the accumulation of mutations in the genome under various stress conditions that eventually aids in superior adaptability of the pathogen in the host.

## Supporting information

**S1 Fig. Impact of in vitro stress conditions. (A)** Oxidative stress; **(B)** Nitrosative stress; and **(C)** hypoxic stress experiments were performed as described in Methods. CFUs were enumerated at indicated time points. **(D)** Competition experiment was performed in hypoxic condition by mixing *Rv* and *RvΔdKO* in equal ratio (1,1). CFUs were enumerated on kanamycin and hygromycin containing 7H11-OADC plates. Graphpad Prism software was used for statistical analysis. Data represents mean and standard deviation.
(TIF)

**S2 Fig. Mutation rate analysis. (A).** WGS of Rv grown in vitro (n = 3) compared with the reference genome of H37Rv (NCBI). Heat map showing the percentage of SNPs accumulated in the Rv. Blosum score provide information about the synonymous and non-synonymous changes due to SNP. Percentage mutation provides information about the percent of strains sequenced where a particular SNP is detected. (**B**) Mutation per million bp was calculated by: sum of all the mutations obtained for a in all the sequenced samples /(4.4 x No. of samples sequencing/strain). Graph shows mutation spectrum in the laboratory Rv strain grown in vitro in comparison with the reference genome. (**C**) Graph shows mutation spectrum of RvΔung, RvΔudgB and RvΔdKO and strains grown in vitro in comparison with Rv strain. (**D**) 50,000 cells per ml were used for performing mutation rate experiment. Graph represents log

values of CFUs enumerated at 0 and 15th day. (**E**) Representative images of colonies of Rv, RvΔung, RvΔudgB and RvΔdKO on rifampicin plates.
(TIF)

**S3 Fig. Sequencing of RRDR.** 383 bp region of rpoB was PCR amplified that includes 81 bp RRDR using rpoB forward and reverse primers using genomic DNA obtained from rifampicin resistant colonies of Rv, RvΔung, RvΔudgB and RvΔdKO. Sequence alignment was performed using Cluster Omega software and only higher than 5 SNP per position was selected. Nucleotide mutation, frequency of occurrence of mutation and corresponding change in amino acid is shown from 419–491 positions.
(TIF)

**S4 Fig. Guinea pig infection. (A)** Representative images of lungs and spleen isolated from Rv, RvΔung, RvΔudgB and RvΔdKO infected guinea pigs. (**B**) CFU analysis at day 1 (n = 3 per strain) and 56 days post infection (n = 5 per strain). Graph pad software was used for performing statistical analysis (Unparied t-test). Data represents mean and standard deviation. $^{*}$ $p<0.05$. (**C**) Heat map showing the synonymous SNPs accumulated in Rv, RvΔung, RvΔudgB, RvΔdKO isolated from guinea pig lungs compared with the sequence of Rv grown in vitro. Percentage mutation provides information about the percent of strains sequenced where a particular SNP is detected. (**D**) Non-sense/non-synonymous mutations identified in the RvΔdKO isolated from guinea pig lungs. (**E**) Ciprofloxacin MIC determination of Rv, RvΔung, RvΔudgB and RvΔdKO. **f**) Gross pathology of lungs and spleen isolated from guinea pigs infected with Rv +RvΔdKO or Rv +RvΔdKO::ung-udgB.
(TIF)

**S5 Fig. Survival of mutant strains ex vivo. (A-c)** Single cell suspension of Rv, RvΔung, RvΔudgB and RvΔdKO was used for the infection in the murine cell line (**A**). RAW 264.7; (**B**) THP1; (**C**) activated RAW 264.7 cells and, (**D**) peritoneal macrophages. CFUs were enumerated at indicated time points on 7H11-OADC containing plates. (**E**) Competition experiment in THP1 was performed by mixing Rv and RvΔdKO in 1:1 ratio. CFUs were enumerated on kanamycin and hygromycin containing plates. Graph pad software was used for performing statistical analysis using unpaired t-test. Data represents mean and standard deviation. $^{**}$ $p<0.005$.
(TIF)

**S1 Text. Growth kinetics and survival under in vitro stress conditions.**
(DOCX)

**S2 Text.**
(DOCX)

**S3 Text. Sequencing of RRDR.**
(DOCX)

**S4 Text.**
(DOCX)

**S5 Text. Survival of mutants ex vivo.**
(DOCX)

**S1 Table. Comparison of Rv in vitro with reference Rv (NCBI).**
(XLSX)

**S2 Table. Mutation Spectrum of Rv in vitro.**
(DOCX)

**S3 Table. Comparison of Rv in vitro with RvΔung in vitro, RvΔudgB in vitro and RvΔdKO in vitro.**
(XLSX)

**S4 Table. Mutation spectrum of RvΔung in vitro, RvΔudgB in vitro and RvΔdKO in vitro.**
(DOCX)

**S5 Table. Comparison of Rv in vitro with Rv (G.P) isolated from guinea pig lungs.**
(XLSX)

**S6 Table. Mutation Spectrum of Rv (GP).**
(DOCX)

**S7 Table. Comparison of Rv in vitro with Rv (GP), RvΔung (GP), RvΔudgB (GP) and RvΔdKO (GP).**
(XLSX)

**S8 Table. Mutation spectrum of Rv(GP), RvΔung(GP), RvΔudgB(GP) and RvΔdKO(GP).**
(DOCX)

**S9 Table. Comparison of RvΔdKO in vitro with RvΔdKO ciprofloxacin resistant (CR) strains.**
(XLSX)

**S10 Table. Mutation spectrum of RvΔdKO in vitro and RvΔdKO ciprofloxacin resistant strains.**
(DOCX)

**S11 Table. List of DNA oligomers used in the study.**
(DOCX)

**S12 Table. Source Data File.**
(XLSX)

## Acknowledgments

We thank the Tuberculosis Aerosol Challenge Facility at ICGEB animal facility and staff for their help in performing animal infection experiments. We would like to thank the bio-containment facility (BSL3) at NII. Vectors (pYUB1471, pYUB854) and mycobacterium phage (phAE159) are a kind gift from Prof. William R. Jacobs's laboratory. We thank Dr. Swati Saha for critically reading the manuscript.

## Author Contributions

**Conceptualization:** Saba Naz, Umesh Varshney, Vinay Kumar Nandicoori.

**Data curation:** Saba Naz, Shruti Dabral, Dhiraj Kumar.

**Formal analysis:** Saba Naz, Shruti Dabral, Dhiraj Kumar.

**Funding acquisition:** Umesh Varshney, Vinay Kumar Nandicoori.

**Investigation:** Saba Naz, Sathya Narayanan Nagarajan, Divya Arora, Lakshya Veer Singh, Pradeep Kumar.

**Methodology:** Yogendra Singh, Umesh Varshney, Vinay Kumar Nandicoori.

**Project administration:** Umesh Varshney, Vinay Kumar Nandicoori.

**Supervision:** Yogendra Singh, Umesh Varshney, Vinay Kumar Nandicoori.

**Validation:** Saba Naz.

**Visualization:** Saba Naz.

**Writing – original draft:** Saba Naz, Vinay Kumar Nandicoori.

**Writing – review & editing:** Saba Naz, Umesh Varshney, Vinay Kumar Nandicoori.

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
