## [Decision Letter · Decision Letter 0]

9 Nov 2020

Dear Dr. Nandicoori,

Thank you very much for submitting your manuscript "Compromised base excision repair pathway in Mycobacterium tuberculosis imparts superior adaptability in the host" for consideration at PLOS Pathogens. As with all papers reviewed by the journal, your manuscript was reviewed by members of the editorial board and by several independent reviewers. In light of the reviews (below this email), we would like to invite the resubmission of a significantly-revised version that takes into account the reviewers' comments.

Several of the reviewers' concerns are likely easy to address with the existing information. However, this paper cannot be accepted without the animal data with the complemented mutant. This is essential. The authors need to examine the nature of the mutations that accumulated. The authors need to use non-DNA-damage-inducing drug (not a fluoroquinolone) to determine if UDG deletion can alter mutation rate.

We cannot make any decision about publication until we have seen the revised manuscript and your response to the reviewers' comments. Your revised manuscript is also likely to be sent to reviewers for further evaluation.

Sincerely,

Helena Ingrid Boshoff

Associate Editor

PLOS Pathogens

Sabine Ehrt

Section Editor

PLOS Pathogens

Kasturi Haldar

Editor-in-Chief

PLOS Pathogens

orcid.org/0000-0001-5065-158X

Michael Malim

Editor-in-Chief

PLOS Pathogens

orcid.org/0000-0002-7699-2064

Several of the reviewers' concerns are likely easy to address with the existing information. However, this paper cannot be accepted without the animal data with the complemented mutant. This is essential. The authors need to examine the nature of the mutations that accumulated. The authors need to use non-DNA-damage-inducing drug (not a fluoroquinolone) to determine if UDG deletion can alter mutation rate.

Reviewer's Responses to Questions

**Part I - Summary**

Reviewer #1: The goal of the authors of this paper set forth to establish the identity and functionality of annotated genes udg and ung as uracil glycosylases as well as to examine their role in DNA repair, mutagenesis and virulence in Mtb. The role of DNA repair systems in Mtb pathogenesis and mutagenesis is an important topic that is understudied. To pursue their goals, the authors create null mutants of TB uracil DNA glycosylases (ung and udgB) and construct complemented strains. The phenotypes of these mutants shown are:

- loss of uracil glycosylase activity in lysates.

- No growth defect for mutants under physiological conditions.

- Increase of the mutation rate in mutants under physiological conditions by RifR assay or ciprofloxacin resistance

- The ung deletion increases the mutation frequency during nitrosative stress/ the udgB deletion increases mutation frequency during oxidative stress.

- The deletion of both the genes confers hypervirulence in guinea pig infection as measured by granulomas and bacterial load.

- By WGS, colonies of ΔudgB and double mutant showed a higher number of SNPs compared with Δung and WT.

- The double mutant is more fit in serial passage and competition experiments in macrophage and mouse infection

Overall, the topic is of interest and a new DNA repair pathway important for Mtb pathogenesis is an important finding. However, some problems with data presentations, controls, and interpretation somewhat limit the potential impact of the findings, as presented.

Reviewer #2: The modulation of genomic integrity in Mycobacterium tuberculosis through spontaneous or stress-induced mutagenesis has been hypothesized to be central to the development of drug resistance in this organism. Consequently, the study of DNA repair pathways including nucleotide/base excision repair pathways, together with the activity of mutator polymerases, is expected to provide useful insight into the molecular mechanisms of drug resistance in mycobacteria. In this submission, Naz and colleagues assess the role of base excision repair in mutagenesis and the emergence of drug resistance in M. tuberculosis. The authors generate single and double mutants that lack the Udg and Ung uracil DNA glycosylases (UDGs) and test these in several model systems. The authors confirm that mutation of the corresponding genes results in loss of uracil DNA glycosylase activity in cell free extracts, genetic complementation restored the activity. The mutants did not display any growth defects under standard conditions in broth culture or stress conditions. Whole genome sequencing suggested that loss of these enzymes did not yield increased chromosomal mutations but was associated with increased mutation rate to Rifampin as determined by the Luria-Delbruck fluctuation assay. The mutants also appeared to have increased DNA damage-induced mutation frequencies in the presence of oxidative and nitrosative stress. In the guinea model of tuberculosis infection, the mutants displayed enhanced colonization of lung tissue when compared to the wild type and accumulated more mutations than the wild type. Upon treatment with ciprofloxacin, loss of UDGs was associated with accumulation of mutations in drug targets such as GyrA. In competition experiments, the UDG double mutant out competed the wild type strain. The authors conclude the UDGs are important mediators of fitness/adaptation and drug resistance. Overall, there are some interesting data here but there are also notable concerns.

Reviewer #3: The manuscript by Naz et al describes the characterization of uracil glycosylases mutant strains ung and udgB with regards to growth profiles and mutation spectrum analyses, both in vitro and in vivo. The authors generate deletion strains of ung, udgB, and the double mutant, along with complemented strains for each. They found that the single mutants had slight increases in mutation rates, while the double mutant was up 22-fold (using spontaneous Rif resistance analysis). They also looked at RifR mutation rate in response to nitrositive and oxidative stresses, with differential increased mutation rates among the single and double mutants.

The authors go on to use the guinea pig model of infection to assess the survival and mutability of their uracil glycosylase mutants in vivo. They found that the double mutant (RvΔdKO) showed a ~0.5 log increase in CFU’s (after 56 days), suggesting this strain is more virulent compared to wild type Rv. Using WGS analysis, colonies derived from post-infection of the mutant strains showed higher rates of SNP accumulation for the udgB and RvΔdKO strains, suggesting a link between the hyper virulence of RvΔdKO and the hypermutability of the uracil glycosylase mutants in vivo. Finally, the authors go on to show that the loss of UDG’s accelerated evolution to ciprofloxacin resistance, and that the RvΔdKO has a competitive advantage over Rv in peritoneal macrophages, but only after successive rounds of infection.

Overall, the paper is well written and describes a thorough series of experiments showing how excision repair mutants, incapable of excising uracil from DNA, generate SNPs under stress in vivo. This mutability ultimately gives them a path that allows them to escape host defenses, leading to increased virulence and competitive growth advantages. While such studies have been performed in other pathogens (as discussed by the authors in th Discussion), I have not seen such a thorough analysis of the evolvability of DNA repair mutants in M. tuberculosis before.

Four main questions:

1. One question that is unclear to me, and perhaps the authors could help with this. When they lists the results of mutation spectrums from GWS analyses, are the base changes listed in the figures in tables (e.g., C > T) for one strand of the chromosome? Otherwise, wouldn’t the number of C > T changes always equal the number G > A changes in a genome wide analysis? In other words, what distinguishes a C > T change from a G > A change in Fig. 5e for any CG base pair.

A clarification would help those us who don’t see mutation spectrum analyses often enough.

An explanation on the above question may clarify a related concern regarding the mutation spectrum analysis discussed in the text and shown in Figure 4h. The authors state in lines 205-206 that “A spectrum analysis of mutations showed a clear predisposition towards G>A and C>T mutations…:., but the data show that the increase in mutations in the udgB and RvΔdKO strains are mostly T>C and C>A, mutations not to be expected from loss of uracil glycosylase.

2. Can the authors speculate as to why with a 0.5-fold increase in CFUs with the RvΔdKO stain (Fig. 3c), they see a lower rate of granuloma formation of RvΔdKO relative to the udgB strain?

3. If I understand Fig. 3b correctly, the heat map shows the percentage of colonies picked for any one mutant that contains that SNP. Perhaps unexpectantly, the results show that many of the SNPs are present at 100% in the single mutants (i.e., 4 out of 4), but are totally absent in the double mutant (0%). Is this counterintuitive?

4. Could the authors speculate what other repair genes, if inactivated, might recapitulate their results with the UNG deletion strain. I gather it would include mutations in repair genes that include a low to moderate increase in mutation rate (like ~ 20%). Included in this discussion would be the acknowledgement that some mutants would be too mutagenic to allow evolution to occur (e.g., a proofreading mutation in dnaE1, or the recently described endoMS/nucS mutation in mismatch repair).

**Part II – Major Issues: Key Experiments Required for Acceptance**

Reviewer #1: The finding of a hypervirulent phenotype in a mutant is exciting but should be interpreted cautiously. There are examples of such findings being due to spontaneous attenuating mutations in the wild type stock, which do not carry over to the mutant because of clonal selection in the construction of the mutant. This gives the appearance of hypervirulence. This issue is solved by in vivo testing of a complemented strain, which should reverse the hypervirulence.

A major conclusion of the paper is that accumulation of mutations in UDG mutants confers fitness during infection due to enhanced evolutionary traits, but this interpretation is difficult to prove. For example, it could be that uracil excision is lethal in the setting of high uracil load and therefore the absence of the repair system provides an advantage.

The assertion that these strains can be used to screen for targets of drugs isn’t convincing and doesn’t add to the story that is presented (ie much of Figure 5). The mutations are not functionally verified and cannot be directly linked to drug action. This line of investigation would be best excluded from the paper to instead focus on the other phenotypes of the mutants.

Reviewer #2: 1. Mutation frequency experiments in figure 3 F&G appear to be done on select strains for select assays. Why are all strains not shown on this grapf? In addition, genetically complemented strains are missing . Why is this the case?

2. For the guinea pig infections, it is unclear how the data on numbers of granulomas was determined. There is not sufficient detail in the methods section nor the figure legend to determine this. It seems difficult to count granulomas by just looking at the surface of excised lungs. This would require sectioning and some form of morphometry on serial sections. Moreover, the gross morphology shown in Figure 4A is unconvincing. It is not immediately apparent that there is a difference between the wild type and the mutants. Furthermore, data on the genetically complemented strains is not presented for any of the animal data. This significantly weakens the study

3. What is the purpose of showing the genes wherein mutations accumulated during guinea infection? It seems that the more interesting data here would be the nature of the mutation and if there were any hotpots in drug resistance conferring loci. The rest of the information is not useful.

4. Why was ciprofloxacin used to determine if UDG deletion can accelerate mutation rate? The choice of this drug needs to be rationalized as treatment with ciprofloxacin would have overlapping effects with loss of UDGs in the mutant due to the induction of a DNA-damage response and the possible induction of the activity of mutator polymerases. Perhaps this point is illustrated by the very high mutation rates reported for the fluoroquinolone used. Using a drug that induces a DNA damage response does not make intuitive sense when trying to determine the mutator effect of UDG loss. This should be attempted with another drug that does not affect DNA metabolism

5. It is unclear what the authors are trying to illustrate with the competition experiment as the result is not sufficiently resolved. The difference between the double mutant and wild type is convincing but why does this happen in competition and not in individual culture? What is the mechanism?

Reviewer #3: No additional experiments required.

**Part III – Minor Issues: Editorial and Data Presentation Modifications**

Reviewer #1: 1) The Supplementary figures are missing from the SI file (only legends and tables are present).

2) Although the verification of the mutant genotypes is convincing and done by multiple methods, some of the PCR verification could be improved (for example 1B, F2-R2 WT is not clear)

3) In Figure 2b, why is the inoculum in Sauton’s medium so high?

4) Figure 2e: a part of the glycosylase activity is due to ung (decreased product formation in the ung mutant) but there is no activity at all in the udgB single mutant (where is the ung-dependent activity?): how do the authors explain this result?

5) The mutation spectrum in mutant vs WT in Rif experiments is not reported. They also should regroup the data so that the same mutations are together (e.g.G>A=C>T)

6) How do the authors explain that there is no increase in spontaneous mutations observed using WGS but they detect an increase with rifR and CipR resistance as a readout?

7) The text states that there is a “Predisposition towards G>A and C>T mutations” (Figure 4h: lines 205-208), but the data in the figure does not support this, those mutation types are no more common than C>A or T>C.

8) Competition experiments: in macrophages (figure 6c) is accomplished by screening based on HygR/S. Is it possible some of the HygR colonies are due to chromosomal mutagenesis rather than present of the HygR marker given the mutagenic phenotypes of the strains?

9) In the mouse experiments (Figure 6e): the data is difficult to understand and the phenotype is weak. What is the Y axis here? This data should present the result as in Figure 6c (proportion of each strain in the mix). Optimally, this result should also include a control of WT-hyg/WT-kan competition to control for differences in plating efficiency between the markers.

Other Comments:

1) The text states “2 biological replicates in triplicate” Were statistical tests done with n=6 or n=2?

2) Fig 2d and e are not properly labeled to differentiate the samples incubated with/ without Ugi.

3) 2e) Is there a reason for the low level of product when �udgB is complemented? Is the protein being under expressed?

4) “Next we analyzed the 159 unique SNPs that were found in RvΔung, RvΔudgB, and RvΔdKO in comparison with Rv in vitro (Figs. 3c)” Should be 3b.

5) “To examine the acquisition of spontaneous mutations in the absence of ung, udgB, or both, we performed mutation rate analysis using a fluctuation test (Figs. 2c-d)” This refers to 3c.

6) “Collectively the data suggests that UDG mutants were hypermutable (Figs. 2d-e)” – Also think this is wrong. Should be 3d-e

7) 6b and 6C: The number of replicates used is not given.

8) Figure 6e does not have Y axis label.

9) What is the statistical test used in all panels? It should be given in each legend or explained in a specific paragraph in material and method (only given for figure 3d).

10) The global organization of the paper could be improved. We suggest:

a. Spontaneous mutation (WGS, rif, cip)

b. Stress- and infection- induced mutation

c. Hyper infection phenotypes (macrophage, mice, guinea pig)

Reviewer #2: Figure 1A-E: It is unclear what the authors intend to show here. Without the sizes of the expected PCR product, or the sizes of the markers, the illustrations and gels are not useful to determine the genetic integrity of the strains used in the study. Whilst some sizes are indicated in the legend it is difficult to follow. Furthermore, Southern Blots are the more acceptable way of confirming genetic mutants

Figure 4C is referred to in the text prior to Figure 4A and B.

Figure 5 A should be removed and given as MIC values in the text.

Reviewer #3: Minor points:

1. Authors should report the concentrations of antibiotics used in the study.

2. Line 159-160. Figure cited should be Fig. 3b, not 3c.

3. Lines 167 and 173. Figures cited should be Fig. 3c-d, not Fig. 2 c-d.

4. Line 200: Since the authors are discussing the vivo results, I think they mean Fig. 4 (e-f).

5. Line 202: …..showed significant SNP accumulation in UDG mutant strains.

Authors should note the figure that shows this (i.e, Fig. 4g).

6. Line 232: The authors should use the 3 letter codes for amino acids here (Asp>Asn or Gly), to prevent confusion when talking about A>G or G>A base pair changes in the same sentence.

PLOS authors have the option to publish the peer review history of their article (what does this mean?). If published, this will include your full peer review and any attached files.

Reviewer #1: No

Reviewer #2: No

Reviewer #3: No
---

## [Editor Report · Decision Letter 1]

4 Mar 2021

Dear Dr. Nandicoori,

We are pleased to inform you that your manuscript 'Compromised base excision repair pathway in Mycobacterium tuberculosis imparts superior adaptability in the host' has been provisionally accepted for publication in PLOS Pathogens.

Best regards,

Helena Ingrid Boshoff

Associate Editor

PLOS Pathogens

Sabine Ehrt

Section Editor

PLOS Pathogens

Kasturi Haldar

Editor-in-Chief

PLOS Pathogens

orcid.org/0000-0001-5065-158X

Michael Malim

Editor-in-Chief

PLOS Pathogens

orcid.org/0000-0002-7699-2064

The authors have addressed the concerns of the reviewers and performed the additional critical experiments (including infections and competition assays with the complemented control strain) to establish that the virulence and competitive advantage of the double knockout mutant are indeed due to loss of the two glycosylases.
---

## [Editor Report · Acceptance letter]

16 Mar 2021

Dear Dr. Nandicoori,

We are delighted to inform you that your manuscript, "Compromised base excision repair pathway in Mycobacterium tuberculosis imparts superior adaptability in the host," has been formally accepted for publication in PLOS Pathogens.

Best regards,

Kasturi Haldar

Editor-in-Chief

PLOS Pathogens

orcid.org/0000-0001-5065-158X

Michael Malim

Editor-in-Chief

PLOS Pathogens

orcid.org/0000-0002-7699-2064